# Probabilistic reconciliation of mixed-type hierarchical time series

**Lorenzo Zambon**[1]    **Dario Azzimonti**[1]    **Nicolò Rubattu**[1]    **Giorgio Corani**[1]

[1]Dalle Molle Institute for Artifcial Intelligence (IDSIA), USI-SUPSI, Lugano, Switzerland

## Abstract

Hierarchical time series are collections of time series that are formed via aggregation, and thus adhere to some linear constraints. The forecasts for hierarchical time series should be *coherent*, i.e., they should satisfy the same constraints. In a probabilistic setting, forecasts are in the form of predictive distributions. Probabilistic reconciliation adjusts the predictive distributions, yielding a joint reconciled distribution that assigns positive probability only to coherent forecasts. There are methods for the reconciliation of hierarchies containing only Gaussian or only discrete predictive distributions; instead, the reconciliation of mixed hierarchies, i.e. mixtures of discrete and continuous time series, is still an open problem. We propose two different approaches to address this problem: *mixed conditioning* and *top-down conditioning*. We discuss their properties and we present experiments with datasets containing up to thousands of time series.

## 1 INTRODUCTION

Hierarchical time series are collections of time series formed via aggregation. For example, the aggregation of the regional levels of tourism yields the national level of tourism; the aggregation of the sales of individual items yields the sales of a group of items, and so on. Forecasts for hierarchical time series should be *coherent*; for instance, the sum of the forecasts of the regional tourism levels should match the forecast for the national tourism level.

Hierarchical forecasts are usually generated in two steps. First, incoherent forecasts are independently generated for each time series (*base forecasts*). Then, they are adjusted to become coherent (*reconciliation*). Reconciled forecasts, besides being coherent, are generally more accurate than the base forecasts: indeed, forecast reconciliation is a spe-

cial case of forecast combination [Hollyman et al., 2021]. Athanasopoulos et al. [2024] provides a review of methodologies and applications of forecast reconciliation.

Most methods [Hyndman et al., 2011, Han et al., 2021, Di Fonzo and Girolimetto, 2024] only reconcile the point forecasts. The state-of-the-art method is minT [Wickramasuriya et al., 2019], whose coherent forecasts minimize the expected mean squared error. However, *reconciled predictive distributions* are needed [Kolassa, 2023] to support decision making. A principled definition of probabilistic reconciliation was given by Panagiotelis et al. [2023]. The probabilistic reconciliation of Gaussian base forecasts and its relation with minT have been studied by Corani et al. [2020], Wickramasuriya [2023], while the probabilistic reconciliation of forecasts for count time series has been studied by Corani et al. [2024], Zambon et al. [2024b,a]. An alternative research line is constituted by end-to-end models [Rangapuram et al., 2021, 2023, Olivares et al., 2024, Das et al., 2023], which produce coherent forecasts for the entire hierarchy.

An open problem is, however, the reconciliation of *mixed hierarchies*, whose disaggregated time series have low-count values, while the aggregated ones are smooth and thus modelled as continuous. This situation is for instance common in retail [Boylan and Syntetos, 2021, Chap. 6.8] and there are currently no suitable methods for this case: "*the development of algorithms to handle mixtures of discrete and continuous data [...] represents a bold research agenda*" [Athanasopoulos et al., 2024].

We propose two approaches for the probabilistic reconciliation of mixed hierarchies. The first (*mixed conditioning*) adopts reconciliation via conditioning [Corani et al., 2020, Zambon et al., 2024b]. It creates a mixed joint distribution of all the base forecasts, where the bottom are defined over counts and the upper over real numbers. The joint predictive distribution is then conditioned on the hierarchy constraints, yielding a coherent reconciled distribution that only includes coherent forecasts. This approach is theoreti-

cally well-grounded but, for reasons discussed later, is not suitable for large hierarchies.

We thus propose also a second approach, which we call *top-down conditioning*. It works in two steps: first, the upper base forecasts are reconciled via conditioning, using only the hierarchical constraints between the upper; then, the bottom distributions are updated in a probabilistic top-down fashion. We show that this approach successfully reconciles hierarchies containing thousands of intermittent bottom time series, taken from the M5 competition Makridakis et al. [2022].

The paper is organized as follows. In Sec. 2, we discuss hierarchical forecasting and probabilistic reconciliation. In Sec. 3, we show how to reconcile mixed hierarchies via conditioning, while in Sec. 4 we present the top-down conditioning approach. We present experiments on real datasets in Sec. 5. The conclusions are in Sec. 6. We provide the proofs in the appendix.

## 2 HIERARCHICAL FORECASTING

Hierarchical time series are collections of time series that are formed via aggregation and therefore satisfy some summing constraints. For instance, in Fig. 1, the time series $u_1$ is equal to the sum of the time series $u_2$ and $u_3$, and so on. The lowest level of the hierarchy contains the *bottom time series*, which are denoted by $\boldsymbol{b} = [b_1, \ldots, b_m]^T$. All the remaining time series are referred to as aggregated or *upper time series*, and are denoted by $\boldsymbol{u} = [u_1, \ldots, u_k]^T$. Finally, we denote by $\mathbf{y} = \left[\boldsymbol{u}^T, \boldsymbol{b}^T\right]^T \in \mathbb{R}^n$ the vector of all the time series. For simplicity, we do not show the time index. The hierarchy constraints are expressed as:

$$\mathbf{y} = \mathbf{S}\boldsymbol{b}, \quad \text{with } \mathbf{S} = \begin{bmatrix} \boldsymbol{A} \\ \hline \mathbf{I} \end{bmatrix}, \tag{1}$$

where $\mathbf{I} \in \mathbb{R}^{m \times m}$ is the identity matrix. $\mathbf{S} \in \mathbb{R}^{n \times m}$ is called *summing matrix* and $\boldsymbol{A} \in \mathbb{R}^{k \times m}$ *aggregating matrix*. For the hierarchy of Fig. 1, we have:

$$\boldsymbol{A} = \begin{bmatrix} 1 & 1 & 1 & 1 \\ 1 & 1 & 0 & 0 \\ 0 & 0 & 1 & 1 \end{bmatrix}.$$

The *base forecasts* are the univariate forecasts produced independently for each time series. In this work, we assume the base forecasts to be in the form of predictive distributions. We denote by $\hat{\pi}_B$ and by $\hat{\pi}_U$ the base forecast distributions for the bottom and upper time series and by $\hat{\pi}$ the base forecast distribution for the entire hierarchy. Depending on the context, $\pi$ denotes either a probability mass function or a density.

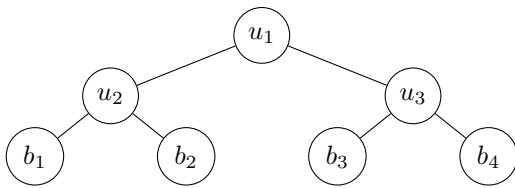

Figure 1: A hierarchy with 4 bottom and 3 upper time series.

**Probabilistic reconciliation.** Let us introduce the *coherent* subspace $\mathcal{S} := \{\mathbf{y} \in \mathbb{R}^n : \mathbf{y} = \mathbf{S}\boldsymbol{b}\}$, which is the set of points that satisfy the hierarchical constraints. The base forecast distribution is *incoherent*, since its support is not contained in $\mathcal{S}$. The aim of probabilistic reconciliation is to find a joint *reconciled distribution* $\widetilde{\pi}$ that gives positive probability only to the points of $\mathcal{S}$. Note that it is sufficient to compute the reconciled distribution $\widetilde{\pi}_B$ for the bottom time series. Indeed, the the reconciled distribution $\widetilde{\pi}$ on the entire hierarchy is then obtained by extending $\widetilde{\pi}_B$ in a coherent way:

$$\widetilde{\pi}(\boldsymbol{u}, \boldsymbol{b}) = \widetilde{\pi}_B(\boldsymbol{b}) \, \mathbb{1}_{\boldsymbol{u}=\boldsymbol{Ab}}, \tag{2}$$

where $\mathbb{1}$ is 1 if $\boldsymbol{u} = \boldsymbol{Ab}$ and 0 otherwise. In the following, we will thus only show how to compute $\widetilde{\pi}_B(\boldsymbol{b})$.

**Probabilistic bottom-up.** The simplest reconciliation approach is the *probabilistic bottom-up*, obtained by setting:

$$\widetilde{\pi}_B = \hat{\pi}_B.$$

The probabilistic bottom-up simply ignores the base forecasts of the upper time series, and has therefore limited accuracy. The marginal distribution of the upper time series reconciled via bottom-up is given by:

$$\hat{\pi}_{bu}(\boldsymbol{u}) := \sum_{\boldsymbol{b}: \, \boldsymbol{Ab}=\boldsymbol{u}} \hat{\pi}_B(\boldsymbol{b}). \tag{3}$$

**Reconciliation via conditioning.** Reconciliation via conditioning conditions the incoherent distribution $\hat{\pi}$ on the hierarchy constraints. If $\hat{\pi}$ is discrete, the reconciled distribution is given by [Zambon et al., 2024b]:

$$\widetilde{\pi}_B(\boldsymbol{b}) := Prob\left(\hat{\boldsymbol{B}} = \boldsymbol{b} \mid \hat{\boldsymbol{U}} - \boldsymbol{A}\hat{\boldsymbol{B}} = 0\right)$$
$$\propto \hat{\pi}(\boldsymbol{Ab}, \boldsymbol{b}). \tag{4}$$

Also in the continuous case [Zambon et al., 2024b], it can be shown that $\widetilde{\pi}_B(\boldsymbol{b}) \propto \hat{\pi}(\boldsymbol{Ab}, \boldsymbol{b})$.

Reconciliation via conditioning can be interpreted in a Bayesian way [Corani et al., 2024]. The distribution of the bottom-up reconciliation constitutes the prior. It is then updated to incorporate the information contained in $\hat{\pi}_U$. It treats the forecast $\hat{\pi}_U$ as a virtual evidence [Darwiche, 2009, Ch 3.6], which increases our belief in certain values of the upper time series. The outcome of the updating is $\widetilde{\pi}_U(\boldsymbol{u})$, which is a compromise [Corani et al., 2024, Zambon et al., 2024a] between $\hat{\pi}_{bu}(\boldsymbol{u})$ and $\hat{\pi}_U(\boldsymbol{u})$.

# 3 MIXED RECONCILIATION VIA CONDITIONING

Let us now consider a *mixed hierarchy*, where the bottom time series have low-count values, while the upper ones are smooth and thus treated as continuous. We thus assume that the predictive distributions of the bottom time series are discrete and the predictive distributions of the upper time series are continuous. Also in the mixed case, reconciliation via conditioning is given by Eq. (4).

**Proposition 1.** *For a hierarchy with discrete bottom forecast distributions and continuous upper forecast distributions, the distribution of the bottom time series reconciled via conditioning is:*

$$\widetilde{\pi}_B(\boldsymbol{b}) \propto \widehat{\pi}(\boldsymbol{Ab}, \boldsymbol{b}). \tag{5}$$

We prove Prop. 1 in App. A, extending the results of Zambon et al. [2024b] to the case of mixed-type variables. Here, we assume for simplicity that all the bottom distributions are discrete and all the upper ones are continuous. The treatment of hierarchies with both continuous and discrete time series on the same level is beyond the scope of this paper.

**Sampling from the reconciled distribution.** In the case of mixed hierarchies, the reconciled distribution in Eq. (5) is only available through samples. Our approach is based on importance sampling: we first sample from the bottom base forecast distribution $\widehat{\pi}_B$, and then compute the weights using the upper base forecast distribution $\widehat{\pi}_U$, which is a multivariate Gaussian in our experiments. Here we cannot use the BUIS algorithm [Zambon et al., 2024b] because we have a joint base distribution on the upper time series.

**Minimal example.** We now reconcile a mixed hierarchy with one upper variable ($U$) and two bottom variables ($B_1$ and $B_2$) with base forecasts:

$$\widehat{\pi}_{B_1} = \text{Poisson}(15), \qquad \widehat{\pi}_{B_2} = \text{Poisson}(15),$$
$$\widehat{\pi}_U = \mathcal{N}(40, 5^2).$$

The base, bottom-up, and reconciled distributions for $U$ are shown in Fig. 2. Since the base forecasts have a positive incoherence (40-15-15 > 0), reconciliation increases the means of the bottom distributions (from 15 to 17.8) and decreases the mean of $U$ (from 40 to 35.6). Reconciliation also reduces the variance of the predictive distributions: the variance of $\widehat{\pi}_{bu}$, $\widehat{\pi}_U$, and $\widetilde{\pi}_U$ are respectively 30.1, 25 and 14.9. This intuitive behavior is consistent with the theoretical properties of the Gaussian reconciliation [Zambon et al., 2024a]. Notice that $\widetilde{\pi}_U$ is discrete, as it is obtained by summing the discrete samples of the reconciled bottom distributions.

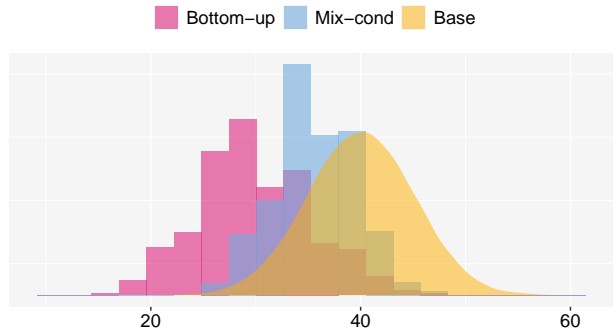

Figure 2: Predictive distribution of $U$: bottom-up (discrete), reconciled (mix-cond, discrete) and base (Gaussian).

**Shortcomings in high dimensions.** We assume the independence of the *discrete* predictive distributions when creating the joint distribution $\widehat{\pi}$. This approach is commonly used given the lack of standard methods, in the *discrete* case, to obtain a multivariate predictive distribution from the marginals. This assumption is viable with a moderate number of bottom time series, but in high dimension, $\widehat{\pi}_{bu}$ is both too peaked and biased. It is too peaked because of the independence assumptions, which leads to an overconfident joint distribution. It is biased because it is obtained by summing many base forecasts: even for the best algorithms, the base forecasts for intermittent time series are biased [Svetunkov and Boylan, 2023]. We noticed in particular that they tend to be overestimated. If the bottom-up distribution is unreliable, the distribution reconciled via conditioning can be worse than the base forecast.

In Fig. 3, we show the distribution of the top level of a hierarchy with 3049 discrete bottom time series, taken from the *M5* competition [Makridakis et al., 2022]. Because of the overestimation bias, the mean of the bottom-up distribution ($\widehat{\pi}_{bu}$, purple) is much larger than the actual value (black triangle) and has a long right tail. It is also more peaked

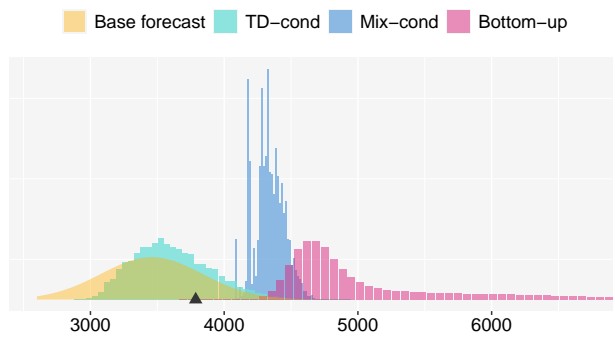

Figure 3: Predictive distribution for the top time series (*M5* dataset, store *WI_1*): base ($\widehat{\pi}_U$, Gaussian), reconciled (TD-cond and Mix-cond), and bottom-up ($\widehat{\pi}_{bu}$, discrete). The black triangle is the actual value.

than the base upper forecast ($\widehat{\pi}_U$, yellow). Notice also the large incoherence between $\widehat{\pi}_{bu}$ and $\widehat{\pi}_U$. Hence, while the *marginal* distributions of the bottom time series are good, the *joint* distribution is unreliable, and therefore also the bottom-up distribution $\widehat{\pi}_{bu}$. Reconciliation via conditioning (Mix-cond), blue in Fig. 3, thus worsens the base forecasts. *Top-down conditioning* (TD-cond), which we discuss in the next section, addresses this problem. It is shown in light blue in Fig. 3, where it provides a sensible reconciled distribution.

## 4 RECONCILIATION VIA TOP-DOWN CONDITIONING

*Top-down conditioning* is a probabilistic reconciliation approach that works in two steps: first, the upper base forecasts are reconciled via conditioning, using only the hierarchical constraints between the upper variables; then, the bottom distributions are updated via a probabilistic top-down procedure. Elgavish [2023] presents a similar idea for the Gaussian case, where it admits an analytical solution. We extend this idea to count time series. First, we formally define the reconciled distribution via top-down conditioning and we study its properties. We show that, in the case of hierarchies with only one upper, the reconciled upper distribution is exactly the base distribution; if there is more than one upper, it is given by the upper distribution partially reconciled via conditioning. We then introduce an algorithm to efficiently sample from the reconciled distribution.

We only consider strictly hierarchical structures, i.e., hierarchies represented by a tree. Moreover, we assume that the hierarchy is *balanced* [Di Fonzo and Girolimetto, 2024]; a precise definition and its implications are given in App. B.1.

Throughout the section we assume that the support of the upper forecast distribution is included in the support of the bottom-up distribution. If all bottom forecast distributions are discrete, then the bottom-up forecast has support on the natural numbers; we thus assume $\widehat{\pi}_U$ to be discrete. In practice, when we run the experiments, the samples from the upper distribution are truncated and rounded before applying the algorithm. We report all the proofs in App. B.

### 4.1 HIERARCHY WITH A SINGLE UPPER

Let us first consider a hierarchy with $m$ bottom and one upper, where $\widehat{\pi}_1, \ldots, \widehat{\pi}_m, \widehat{\pi}_U$ are the base distributions.

**Definition 1.** *Assume that the base forecasts for all the bottom and the upper time series are conditionally independent. We define the reconciled distribution via top-down conditioning as*

$$\widetilde{\pi}_{\mathrm{TD}}(\boldsymbol{b}) := \widehat{\pi}_1(b_1) \ldots \widehat{\pi}_m(b_m) \frac{\widehat{\pi}_U(b_1 + \cdots + b_m)}{\widehat{\pi}_{bu}(b_1 + \cdots + b_m)}. \quad (6)$$

In App. B.2 we show that $\widetilde{\pi}_{\mathrm{TD}}$ is a probability distribution. Note that the bottom-up distribution $\widehat{\pi}_{bu}$, defined in Eq. (3), can be written in this case as

$$\widehat{\pi}_{bu}(u) := \sum_{\substack{b_1, \ldots, b_m: \\ b_1 + \cdots + b_m = u}} \widehat{\pi}_1(b_1) \ldots \widehat{\pi}_m(b_m). \quad (7)$$

Definition 1 could be easily generalized to include the case in which we have a joint distribution $\widehat{\pi}_B$ on the bottom time series; we leave the study of the correlations between discrete bottom forecasts for future work. The reconciled distribution via top-down conditioning has two desirable properties.

**Proposition 2.** $\widetilde{\pi}_{\mathrm{TD}}$ *satisfies the following properties:*

*(i) If* $(\widetilde{B}_1, \ldots, \widetilde{B}_m) \sim \widetilde{\pi}_{\mathrm{TD}} \implies \widetilde{B}_1 + \cdots + \widetilde{B}_m \sim \widehat{\pi}_U$

*(ii) Given* $(\bar{b}_1, \ldots, \bar{b}_m)$ *and* $(\check{b}_1, \ldots, \check{b}_m)$ *such that* $\bar{b}_1 + \cdots + \bar{b}_m = \check{b}_1 + \cdots + \check{b}_m$, *then*

$$\frac{\widetilde{\pi}_{\mathrm{TD}}(\bar{b}_1, \ldots, \bar{b}_m)}{\widetilde{\pi}_{\mathrm{TD}}(\check{b}_1, \ldots, \check{b}_m)} = \frac{\widehat{\pi}_1(\bar{b}_1) \ldots \widehat{\pi}_m(\bar{b}_m)}{\widehat{\pi}_1(\check{b}_1) \ldots \widehat{\pi}_m(\check{b}_m)}$$

The first property explains the name *top-down*: indeed, the reconciled upper distribution is exactly the base upper distribution. The second property specifies how the distribution of the upper is split between the bottom, i.e., proportionally to the base distribution of the bottom.

**Sampling from** $\widetilde{\pi}_{\mathrm{TD}}$. Let us first consider the case $m = 2$. We can rewrite Eq. (6) as

$$\begin{aligned}
\widetilde{\pi}_{\mathrm{TD}}(b_1, b_2) &= \widehat{\pi}_1(b_1) \widehat{\pi}_2(b_2) \frac{\widehat{\pi}_U(b_1 + b_2)}{\widehat{\pi}_{bu}(b_1 + b_2)} \\
&= \sum_u \widehat{\pi}_U(u) \cdot \frac{\widehat{\pi}_1(b_1) \widehat{\pi}_2(b_2)}{\widehat{\pi}_{bu}(u)} \mathbb{1}_{b_1 + b_2 = u} \\
&= \sum_u \widehat{\pi}_U(u) \cdot \widehat{\pi}(b_1, b_2 \,|\, b_1 + b_2 = u). \quad (8)
\end{aligned}$$

Eq. (8) shows that we can sample from $\widetilde{\pi}_{\mathrm{TD}}$ in two steps. First, we sample $u$ from $\widehat{\pi}_U$; then, we sample $(b_1, b_2)$ from the base bottom distribution, conditioned on the constraint $b_1 + b_2 = u$. For the latter step, we introduce Alg. 1. It samples $b_1$ from the marginal distribution of $\widehat{\pi}(b_1, b_2 \,|\, b_1 + b_2 = u)$; $b_2$ is then computed as $u - b_1$.

**Lemma 1.** *The output* $(b_1, b_2)$ *of Alg. 1 is distributed as*

$$\widehat{\pi}(b_1, b_2 \,|\, b_1 + b_2 = u).$$

Analogously, in the general case of $m > 2$, we need to sample from $\widehat{\pi}(b_1, \ldots, b_m \,|\, b_1 + \cdots + b_m = u)$. Since there are $m$ variables and 1 constraint, the direct generalization of Alg. 1 would require sampling from a $(m - 1)$-dimensional joint distribution, which is not feasible when $m$ is large. We thus introduce Alg. 2: the key idea is to iteratively split the

**Algorithm 1** Top-down sampling (2 bottom)

1: **Input**: $\widehat{\pi}_1, \widehat{\pi}_2; u$
2: **Output**: sample $(b_1, b_2)$

3: Define $q_1(b_1) \propto \widehat{\pi}_1(b_1)\,\widehat{\pi}_2(u - b_1)$
4: $b_1 \leftarrow$ sample from $q_1$
5: $b_2 \leftarrow u - b_1$
6: **return** $(b_1, b_2)$

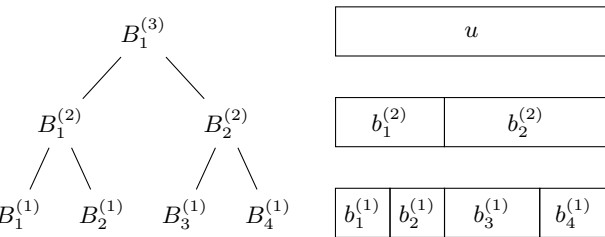

Figure 4: Auxiliary binary tree ($m = 4$, $L = 3$)

variables in two groups, applying each time Alg. 1. For the sake of clarity, we present the algorithm in the case that $m$ is a power of 2; however, we show in App. B.7 that the algorithm can be easily adapted to any $m$. Let us define $\widehat{B}_1 \sim \widehat{\pi}_1, \ldots, \widehat{B}_m \sim \widehat{\pi}_m$, and assume that all the $\widehat{B}_j$'s are independent. We then set $L := \log_2(m) + 1$, and we build an auxiliary binary tree in the following way:

$$
\begin{aligned}
B_j^{(1)} &= \widehat{B}_j, && \text{for } j = 1, \ldots, m, \\
B_j^{(l+1)} &= B_{2j-1}^{(l)} + B_{2j}^{(l)}, && \text{for } l = 1, \ldots, L - 1, \\
&&& j = 1, \ldots, 2^{L-l-1}.
\end{aligned}
$$

An example for $m = 4$ is shown in Fig. 4. For each $l$ and $j$, we denote by $\pi_j^{(l)}$ the distribution of $B_j^{(l)}$. In the first part of Alg. 2, we compute $\pi_j^{(l)}$ for each $j$ and $l$; thanks to the independence assumption, the distribution of the sum is given by the convolution, denoted by $*$:

$$
\pi_j^{(l+1)} = \pi_{2j-1}^{(l)} * \pi_{2j}^{(l)}. \tag{9}
$$

In practice, convolutions can be computed efficiently using the Fast Fourier Transform [Cooley and Tukey, 1965]. In the second part of Alg. 2, we start from $u$ at the top node of the auxiliary binary tree, and proceed downward by iteratively doing conditional sampling using Alg. 1. For example, if $m = 4$, we first draw $(b_1^{(2)}, b_2^{(2)})$ conditioned on $b_1^{(2)} + b_2^{(2)} = u$, then we draw $(b_1^{(1)}, b_2^{(1)})$ conditioned on $b_1^{(1)} + b_2^{(1)} = b_1^{(2)}$ and $(b_3^{(1)}, b_4^{(1)})$ conditioned on $b_3^{(1)} + b_4^{(1)} = b_2^{(2)}$ (Fig. 4).

**Lemma 2.** *The output $(b_1, \ldots, b_m)$ of Alg. 2 is distributed as*

$$
\widehat{\pi}(b_1, \ldots, b_m \mid b_1 + \cdots + b_m = u).
$$

**Algorithm 2** Top-down sampling ($2^{L-1}$ bottom)

1: **Input:** $u; \widehat{\pi}_1, \ldots, \widehat{\pi}_m$
2: **Output:** sample $(b_1, \ldots, b_m)$

3: $L \leftarrow \log_2(m) + 1$
4: ### Compute the $\pi_j^{(l)}$'s
5: $\pi_j^{(1)} \leftarrow \widehat{\pi}_j$ for each $j = 1, \ldots, m$
6: **for** $l = 1, \ldots, L - 1$ **do**
7:     **for** $j = 1, \ldots, 2^{L-l-1}$ **do**
8:       $\pi_j^{(l+1)} \leftarrow \pi_{2j-1}^{(l)} * \pi_{2j}^{(l)}$
9: ### Top-down sampling
10: $b_1^{(L)} \leftarrow u$
11: **for** $l = L - 1, \ldots, 1$ **do**
12:     **for** $j = 1, \ldots, 2^{L-l-1}$ **do**
13:       $\left(b_{2j-1}^{(l)}, b_{2j}^{(l)}\right) \leftarrow$ Alg. 1$\left(\pi_{2j-1}^{(l)}, \pi_{2j}^{(l)}; b_j^{(l+1)}\right)$
14: **return** $\left(b_1^{(1)}, \ldots, b_m^{(1)}\right)$

---

**Algorithm 3** Top-down conditioning (1 upper)

1: **Input:** $\widehat{\pi}_U, \widehat{\pi}_1, \ldots, \widehat{\pi}_m; N$
2: **Output:** sample $\left(b_1^i, \ldots, b_m^i\right)_{i=1,\ldots,m}$

3: **Sample** $\left(u^i\right)_{i=1,\ldots,N} \overset{\text{IID}}{\sim} \widehat{\pi}_U$
4: **for** $i = 1, \ldots, N$ **do**
5:     $\boldsymbol{b}^i \leftarrow$ Alg. 2$\left(u^i; \widehat{\pi}_1, \ldots, \widehat{\pi}_m\right)$
6: **return** $\left(\boldsymbol{b}^i\right)_{i=1,\ldots,N}$

Finally, we introduce Alg. 3 for sampling from the reconciled distribution via top-down conditioning in case of 1 upper and $m$ bottom time series.

**Proposition 3.** *The output of Alg. 3 is distributed as $\widetilde{\pi}_{\mathrm{TD}}$, defined in Eq. (6).*

## 4.2 HIERARCHY WITH $k$ UPPER

Let us now consider the general case of a hierarchy with $m$ bottom and $k$ upper. We generalise Definition 1 as follows.

**Definition 2.** *Let $\widehat{\pi}_1, \ldots, \widehat{\pi}_m$ be the conditionally independent base distributions of the bottom, and $\widehat{\pi}_U$ the multivariate distribution of the upper. We further assume conditional independence between upper and bottom. The reconciled distribution via top-down conditioning is given by*

$$
\widetilde{\pi}_{\mathrm{TD}}(\boldsymbol{b}) := \frac{\widehat{\pi}_1(b_1) \ldots \widehat{\pi}_m(b_m)}{\widehat{\pi}_{bu}(\boldsymbol{Ab})}\, \widehat{\pi}_U(\boldsymbol{Ab}). \tag{10}
$$

We recall that the bottom-up distribution is defined as $\widehat{\pi}_{bu}(\boldsymbol{u}) := \sum_{\boldsymbol{b}:\ \boldsymbol{Ab}=\boldsymbol{u}} \widehat{\pi}_B(\boldsymbol{b})$. This sum is non-empty only if $\boldsymbol{u}$ satisfies the hierarchy constraints. For example, in the case of the hierarchy of Fig. 1, if $u_1 \neq u_2 + u_3$, then $\{\boldsymbol{b}:\ \boldsymbol{Ab} = \boldsymbol{u}\} = \varnothing$, and therefore $\widehat{\pi}_{bu}(\boldsymbol{u}) = 0$.

Since the hierarchy is balanced, we can consider the sub-hierarchy given by only the upper time series (App. B.1). Hence, following the notation of Sec. 2, we can write

$$\boldsymbol{u} = \mathbf{S}^{\mathbf{u}} \boldsymbol{u}^{low}, \qquad (11)$$

where $\boldsymbol{u}^{low}$ is the set of upper time series on the lowest level of the hierarchy. We then denote by $\boldsymbol{u}^{upp}$ the set of all the other upper time series, so that $\boldsymbol{u}^{upp} = \mathbf{A}^{\mathbf{u}} \boldsymbol{u}^{low}$. In the example of Fig. 1, we have $\boldsymbol{u}^{low} = [u_2, u_3]^T$, $\boldsymbol{u}^{upp} = [u_1]$, and $\mathbf{A}^{\mathbf{u}} = [1\ 1]$. Prop. 2 can be generalized as follows.

**Proposition 4.** *The distribution $\widetilde{\pi}_{\mathrm{TD}}$ in Def. 2 satisfies the following properties:*

*(i) If $\widetilde{\boldsymbol{B}} \sim \widetilde{\pi}_{\mathrm{TD}} \implies \boldsymbol{A}\widetilde{\boldsymbol{B}} \sim \widehat{\pi}_U(\boldsymbol{u})\, \mathbb{1}_{\boldsymbol{u}^{upp}=\mathbf{A}^{\mathbf{u}}\boldsymbol{u}^{low}}$*

*(ii) Given $\bar{\mathrm{b}}$ and $\check{\mathrm{b}}$ such that $\boldsymbol{A}\bar{\mathrm{b}} = \boldsymbol{A}\check{\mathrm{b}}$, then*

$$\frac{\widetilde{\pi}_{\mathrm{TD}}(\bar{\mathrm{b}})}{\widetilde{\pi}_{\mathrm{TD}}(\check{\mathrm{b}})} = \frac{\widehat{\pi}_1(\bar{b}_1)\dots\widehat{\pi}_m(\bar{b}_m)}{\widehat{\pi}_1(\check{b}_1)\dots\widehat{\pi}_m(\check{b}_m)}$$

Note that the distribution of $\boldsymbol{A}\widetilde{\boldsymbol{B}}$ in (i) can be written as

$$\widehat{\pi}_U(\mathbf{A}^{\mathbf{u}}\boldsymbol{u}^{low},\ \boldsymbol{u}^{low})\, \mathbb{1}_{\boldsymbol{u}^{upp}=\mathbf{A}^{\mathbf{u}}\boldsymbol{u}^{low}}, \qquad (12)$$

which corresponds to the formula of reconciliation via conditioning (2), with $\boldsymbol{A}, \boldsymbol{u}, \boldsymbol{b}$ replaced by $\mathbf{A}^{\mathbf{u}}, \boldsymbol{u}^{upp}, \boldsymbol{u}^{low}$. This provides the intuition for the algorithm to sample from $\widetilde{\pi}_{\mathrm{TD}}$ in case of hierarchies with more than one upper (Alg. 4). First, we reconcile only the upper forecasts, by conditioning on the constraints between the upper. Note that reconciliation via conditioning is not an arbitrary choice, but is implied by the properties of $\widetilde{\pi}_{\mathrm{TD}}$, as discussed above. If the base forecast for the upper time series is a multivariate Gaussian, reconciliation via conditioning can be done analytically [Corani et al., 2020, Zambon et al., 2024a]. We can then sample from the partially reconciled distribution on the lowest level of the upper, and apply Alg. 2 for each of the lowest upper.

**Proposition 5.** *The output of Alg. 3 is distributed as $\widetilde{\pi}_{\mathrm{TD}}$, defined in Eq. (10).*

---

**Algorithm 4** Top-down conditioning ($k$ upper)

1: **Input:** $\boldsymbol{A}$; $\widehat{\pi}_U, \widehat{\pi}_1, \dots, \widehat{\pi}_m$; $N$
2: **Output:** sample $\left(b_1^i, \dots, b_m^i\right)_{i=1,\dots,m}$

3: $\quad \mathbf{S}^{\mathbf{u}} \leftarrow \text{sub-hier}(\boldsymbol{A})$
4: $\quad \widetilde{\pi}_{U^{low}} \leftarrow \text{cond-reconc}(\mathbf{S}^{\mathbf{u}}, \widehat{\pi}_U)$
5: **Sample** $\left(u_1^i, \dots, u_{k_{low}}^i\right)_{i=1,\dots,N} \overset{\text{IID}}{\sim} \widetilde{\pi}_{U^{low}}$
6: **for** $i = 1, \dots, N$ **do**
7: $\quad$ **for** $j = 1, \dots, k_{low}$ **do**
8: $\quad\quad \bar{\mathrm{b}}^{(j)} \leftarrow \text{Alg. 2}\left(u_j^i; \widehat{\pi}_1, \dots, \widehat{\pi}_m\right)$
9: $\quad b^i \leftarrow \left(\bar{\mathrm{b}}^{(1)}, \dots, \bar{\mathrm{b}}^{(k_{low})}\right)$
10: **return** $\left(b^i\right)_{i=1,\dots,N}$

---

| | $n$ | $m$ | $T$ | $\bar{y}_u$ | $\bar{y}_b$ |
|---|---|---|---|---|---|
| *Syph-small* | 10 | 9 | 209 | 26 | 3 |
| *Syph* | 54 | 53 | 209 | 97 | 2 |
| *M5* | 3060 | 3049 | 1941 | {3448, 718, 387} | 1 |

Table 1: Datasets characteristics: $n$ is the total number of time series, $m$ the number of bottom time series, $T$ the length of the time series , $\bar{y}_b$ and $\bar{y}_u$ the mean of the bottom and upper time series. For *M5*, the three values of $\bar{y}_u$ correspond to the three upper levels of the hierarchy (store, category, department).

## 5  EXPERIMENTS

We consider three different hierarchical datasets. We start with the *Syph* dataset, available from the R package ZIM [Yang et al., 2018]. It provides the *weekly* number of syphilis cases in the US from 2007 to 2010. The hierarchy has 53 bottom time series (one for each state) and one upper time series, the total number of cases in the US. We then consider a reduced version of this dataset (*Syph-small*), which contains only the nine states of the South Atlantic region and their total.

We also consider the high-dimensional dataset of the *M5* competition [Makridakis et al., 2022]. It contains *daily* sales data referring to 10 different stores. The hierarchy of each store has the same structure: 3049 bottom time series (single items) and 11 upper time series, obtained by aggregating the items by department, product category, and store (Fig. 5). We independently reconcile each store and we eventually report the average results.

Tab.1 reports the main characteristics of the datasets. In all datasets, the mean values of the time series justify modelling the bottom time series as counts and the upper time series as continuous.

We always consider the reconciliation of one-step-ahead forecasts. We perform 52 reconciliations on *Syph* and *Syph-small* and 14 reconciliations on *M5* adopting a rolling-origin approach, i.e., at each iteration we increase the time series of one time step, re-compute the base forecasts and reconcile them. The number of reconciled bottom time series is hence 9 x 52 = 486 on *Syph-small*, 53 x 52 = 2756 on *Syph* and 10 (stores) x 3049 (bottom) x 14 = 426'860 on *M5*.

**Methods.** We compute the base forecasts using ADAM [Svetunkov and Boylan, 2023], available from the R package *smooth* [Svetunkov, 2023]. It is a state-space model for probabilistic forecasting of both intermittent and smooth time series. On intermittent time series, it returns the predictive distributions in the form of positive samples. The samples are continuous; we round them to predict count time series, as done by Svetunkov and Boylan [2023]. For

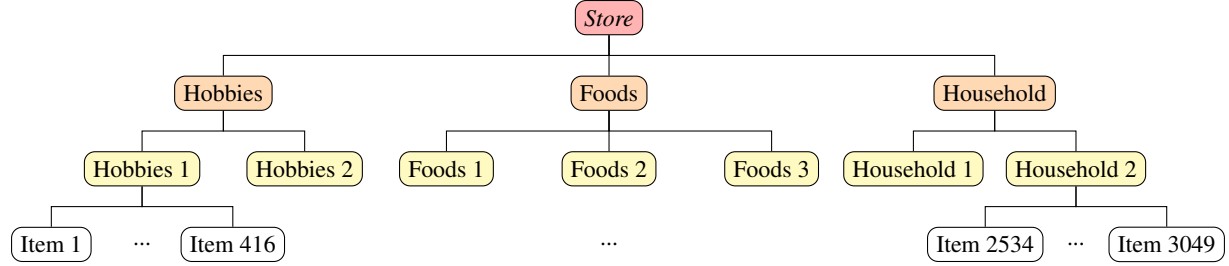

Figure 5: *M5 Store*-hierarchy. The store is in red, categories are in orange, departments are in yellow.

smooth time series, we use ADAM with a Gaussian predictive distribution.

We compare different reconciliation approaches against the base forecast, which is our *baseline*. All methods are implemented in the R package bayesRecon[1] [Azzimonti et al., 2023].

The first is the Gaussian reconciliation (***Gauss***), which we implement as follows. We approximate each bottom base forecast with a Gaussian distribution. We then obtain the joint predictive density for the bottom time series assuming them to be independent. We adopt a multivariate normal as joint density for the upper time series. As in [Corani et al., 2020], its mean is the mean of the base forecasts of the upper time series and its covariance matrix is equal to the covariance of the 1-step-ahead residuals, estimated via shrinkage [Wickramasuriya et al., 2019]. We also assume independence between the forecast of bottom and upper time series. After these approximations, we have a joint Gaussian distribution over bottom and upper time series, which we reconcile analytically following Corani et al. [2020].

We also test a variant of *Gauss*, designed to yield only positive values. We set to zero the reconciled bottom samples that are negative. We then sum up such truncated bottom samples to obtain the reconciled distribution for the entire hierarchy. This approach yields samples that are positive and coherent, but biased. We refer to this as ***Gauss-T***, where the T stands for *truncated*.

The third method is mixed conditioning (***Mix-cond***), which we implement as follows. As for *Gauss*, we model the predictive density of the upper time series as a joint multivariate normal distribution. We then model the bottom joint distribution over counts as the product of discrete distributions and we perform reconciliation as discussed in Sec. 3.

Finally, we consider the top-down conditioning method (**TD-cond**). We adopt the same joint base distribution of *Mix-cond*, but we reconcile it using the methods of Sec. 4. In particular we use Alg. 3 on *Syph-small* and *Syph*, which contain a single upper variable, and Alg. 4 on *M5*, which contains multiple upper variables.

---

[1]The vignette "Reconciliation of M5 hierarchy with mixed-type forecasts" in the package partially reproduces the results.

The implementation of *Mix-cond* and *TD-cond* reconciles a store of the *M5* hierarchy (3060 time series) in a median time of 11.2 (Mix-cond) and 10.9 (TD-cond) seconds on a M1 Mac laptop.

**Indicators.** We assess the point forecasts using the mean scaled absolute error (MASE) [Hyndman, 2006]. Following [Kolassa, 2016], we use the median as point forecast when computing the MASE. We score the 90% prediction intervals using the mean interval score (MIS) [Gneiting, 2011]. We assess the marginal predictive distributions using the ranked probability score (RPS) [Kolassa, 2016] and the joint predictive distribution using the energy score (ES) [Panagiotelis et al., 2023]. We compute RPS and ES using the *scoringRules* R package [Jordan et al., 2019]. We do not compute the ES for the *M5* dataset, since the energy score has computational and sampling issues in high dimensions [Pinson and Tastu, 2013].

We report the improvement over the base forecasts using the skill score values and averaging them across experiments. For instance, the skill score of *Gauss* on ES is:

$$\text{Skill}_\% (\text{ES}, Gauss) = 100 \cdot \frac{\text{ES}(base) - \text{ES}(Gauss)}{(\text{ES}(base) + \text{ES}(Gauss))/2} .$$

A positive skill score implies an improvement with respect to the base forecasts.

**Results.** On *Syph-small*, *Mix-cond* outperforms the other approaches (Tab. 2), as expected in low dimensions. *TD-cond* is not very suitable for this dataset, as it does not exploit the information of the joint bottom-up distribution (which is tenable in this case) to revise the upper base forecast. The Gaussian approaches provide a poor approximation for count distributions and thus they have low performance, especially on the bottom time series. Their positive skill scores on the upper time series are due to a reduction of variance compared to the base forecasts. The *Gauss-T* improves the prediction intervals (scored by MIS) compared to *Gauss*. However, the overall performance of both *Gauss* and *Gauss-T* is generally poor and we no longer comment on them.

On *Syph*, the best-performing approach is either *Mix-cond* or *TD-cond*, depending on the indicator. Besides the average

|  |  | Gauss | Gauss-T | Mix-cond | TD-cond |
|---|---|---|---|---|---|
| ***Syph-small*** |  |  |  |  |  |
| MASE | *Bottom* | -61.7 | -61.7 | **-2.5** | -4.3 |
|  | *Upper* | 13.3 | -2.6 | **23.8** | -0.8 |
| MIS | *Bottom* | -45.4 | -2.8 | **4.4** | -6.3 |
|  | *Upper* | 41.7 | 34.0 | **42.3** | 12.8 |
| RPS | *Bottom* | -56.7 | -52.5 | **4.4** | -3.3 |
|  | *Upper* | 26.3 | 21.3 | **29.0** | 4.9 |
| ES |  | 9.5 | 6.2 | **12.0** | 2.3 |
| ***Syph*** |  |  |  |  |  |
| MASE | *Bottom* | -87.4 | -80.4 | -0.2 | **1.3** |
|  | *Upper* | -8.7 | -62.5 | -5.6 | **0.0** |
| MIS | *Bottom* | -79.9 | -37.0 | 2.9 | **4.4** |
|  | *Upper* | 19.9 | -43.0 | **20.2** | -1.0 |
| RPS | *Bottom* | -91.8 | -88.1 | **17.6** | -4.6 |
|  | *Upper* | 4.4 | -53.4 | **7.0** | 0.0 |
| ES |  | -1.6 | -36.1 | 1.0 | **3.0** |
| ***M5*** |  |  |  |  |  |
| MASE | *Bottom* | -69.1 | -66.7 | 0.5 | **2.8** |
|  | *Upper* | -34.9 | -118.6 | -29.9 | **-0.5** |
| MIS | *Bottom* | -47.8 | -20.3 | 4.9 | **10.6** |
|  | *Upper* | -34.3 | -150.9 | -37.6 | **3.6** |
| RPS | *Bottom* | -55.8 | -50.5 | 8.3 | **14.7** |
|  | *Upper* | -33.2 | -127.8 | -30.0 | **1.3** |

Table 2: Mean skill scores on *Syph-small*, *Syph* and *M5* datasets.

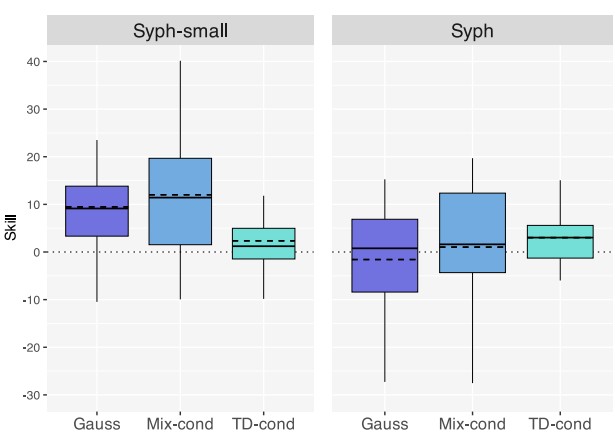

Figure 6: Boxplots of the skill scores on the energy score (ES) on *Syph-small* and *Syph* datasets. The means are shown as dashed lines and the medians as solid lines. Means and medians are different due to the presence of outliers. The Gauss-T method is not shown for reasons of space.

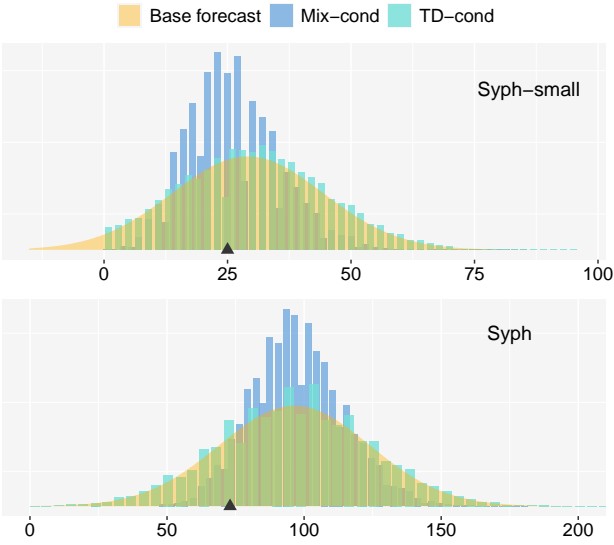

Figure 7: Base forecasts, reconciled distributions (*Mix-cond*, *TD-cond*), and actual values (black triangles) for *Syph-small* and *Syph* dataset.

value of the indicators, it is worth looking at their variability. In Fig.6, we show the boxplot of the skill scores of the ES on the 52 reconciliations of *Syph* and *Syph-small*.

On *Syph-small*, *Mix-cond* yields the highest distribution of the skill scores. The performance of *Gauss* is better on this dataset than on the others since there are few bottom time series, and the Gaussian reconciliation works well on the upper time series. On *Syph*, however, *TD-cond* has both the highest median and the lowest variance of energy score; it is arguably the preferable approach.

In Fig.7 we illustrate the difference between *Mix-cond* and *TD-cond* in two examples of reconciliation. As already pointed out, both approaches return a positive, discrete reconciled distribution for the upper time series. This happens even if the upper base forecasts have a tail of negative values, as in Fig.7. The upper reconciled distribution of *Mix-cond* has lower variance than the base forecasts. This is beneficial in the first example, in which both the bottom-up and the

base forecasts provide valuable information. In the second example, referring to *Syph*, this yields instead a distribution that is peaked around a wrong value.

On *M5*, *Mix-cond* performs poorly (Tab. 2) for the reasons discussed in Sec. 3. *TD-cond*, instead, shows a convincing performance from different viewpoints. On average, it provides a solid improvement on the predictive distribution of the bottom time series and a moderate improvement on the upper time series; the latter is due to the Gaussian reconciliation applied on the 11 upper time series. Moreover, it is more reliable than the competitors, having a much lower

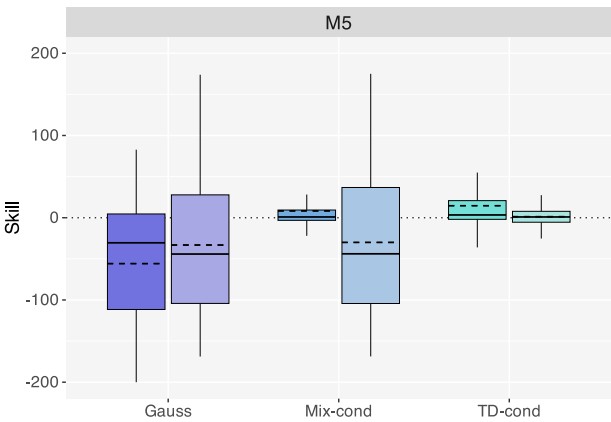

Figure 8: Boxplots of the skill scores on the RPS on *M5* datasets. The means are shown as dashed lines and the medians as solid lines. Means and medians are different due to the presence of outliers. For each method, the left boxplot refers to the bottom, the right one to the upper.

variability of the skill scores on both bottom and upper time series (Fig. 8). Note the different scales of the axes in Fig. 6 and Fig. 8.

## 6   CONCLUSION

We presented two principled methods for the probabilistic reconciliation of mixed-type hierarchical forecasts. *Mixed conditioning* extends previous work on reconciliation via conditioning, but is only effective in moderately-sized hierarchies because of the shortcomings of the bottom-up distribution in high dimensions. Weakening the assumption of conditional independence between the bottom predictive distributions is a promising direction to overcome the problem; we leave this study for future work. The second method is *top-down conditioning*, which can be sensibly used to reconcile large mixed hierarchies. First, the upper forecasts are reconciled via conditioning; in this work, we used Gaussian reconciliation because we assumed the upper forecasts to be jointly Gaussian, but this is not an intrinsic limitation of our method. Then, the bottom forecasts are reconciled via a probabilistic top-down procedure. We introduced top-down conditioning under the assumptions that all the bottom forecasts are discrete and all the upper are Gaussian; moreover, we assumed that the time series are organized into a balanced hierarchy. We leave for future work the extension to more general linearly constrained multiple time series, or to cases in which different types of forecasts are on the same level of the hiearchy.

## Author Contributions

LZ formalized the reconciliation for mixed variables, designed and implemented the conditioning and the top-down algorithms. DA and NR contributed to the formalization of the algorithm and partially developed the code for the experiments and visualization. GC proposed the research topic. All authors substantially contributed to the design of the experiments, the discussion of the results, and the drafting of the manuscript.

## Acknowledgements

Research funded by the Swiss National Science Foundation (grant 200021_212164) and by the Hasler foundation (project: *hierarchical forecasting with mixed hierarchies*).

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

# Appendix

**Lorenzo Zambon**[1]  **Dario Azzimonti**[1]  **Nicolò Rubattu**[1]  **Giorgio Corani**[1]

[1]Dalle Molle Institute for Artifcial Intelligence (IDSIA), USI-SUPSI, Lugano, Switzerland

## A  MIXED CONDITIONING

### A.1  MIXED TYPE DISTRIBUTIONS AND DENSITIES

We recall here some basic notions about distributions and densities, as we need them in Sec. A.2 to derive the formula of reconciliation via conditioning in the mixed case.

A *count* variable $X$ can only assume non-negative integer values; thus it has range $\mathbb{N} = \{0, 1, 2, \dots\}$. The distribution of $X$ is represented by the probability mass function (pmf) $\pi_X$, which assigns a probability to each point of the range. Hence

$$Prob\left(X \in G\right) = \sum_{x \in G} \pi_X(x), \tag{13}$$

for any $G \subset \mathbb{N}$. The pmf is the *density* of $X$ with respect to the counting measure $\mathcal{C}$ over $\mathbb{N}$, defined as $\mathcal{C}(G) = \sum_{j \in \mathbb{N}} \mathbf{1}_{\{j \in G\}}$. Indeed, the sum in Eq. (13) is the integral of the pmf with respect to the measure $\mathcal{C}$ [Billingsley, 2017].

A real-valued random variable $Y$ is *absolutely continuous* (in the following, just *continuous*) if its distribution is absolutely continuous with respect to the Lebesgue measure $\mathcal{L}$. The distribution of $Y$ is then represented by its density $\pi_Y$ with respect to $\mathcal{L}$, such that:

$$Prob\left(Y \in F\right) = \int_F \pi_Y(y) \; dy. \tag{14}$$

for any measurable $F \subset \mathbb{R}$.

We now introduce the mixed case. Let

$$\mathbf{Z} = (X_1, \dots, X_m, Y_1, \dots, Y_k)$$

be a random vector, where the $X_i$'s are discrete and the $Y_j$'s are continuous. We denote by $\pi_{\mathbf{Z}}$ the density of $\mathbf{Z}$ with respect to the product measure $\mathcal{C}^m \otimes \mathcal{L}^k$. Hence, for any measurable $G \subset \mathbb{R}^m$ and $F \subset \mathbb{R}^k$, we have

$$Prob\left(\mathbf{Z} \in G \times F\right) = \sum_{\mathbf{x} \in G} \int_F d\mathbf{y} \; \pi_{\mathbf{Z}}(\mathbf{x}, \mathbf{y}). \tag{15}$$

Note that the sum over $G$ in Eq. (15) is well-posed as $\pi_{\mathbf{Z}}(\mathbf{x}, \mathbf{y}) \neq 0$ only for countably many $\mathbf{x}$'s. See Billingsley [2017] for a detailed discussion of measures and densities.

### A.2  PROOF OF PROPOSITION 1

Let us assume that the forecast distribution for the bottom time series is discrete, while for the upper is continuous. We denote by $\hat{\pi}$ the density of $\widehat{\boldsymbol{Y}} = \left(\widehat{\boldsymbol{U}}, \widehat{\boldsymbol{B}}\right)$ with respect to $\mathcal{L}^k \otimes \mathcal{C}^m$, so that, for any measurable $F \subset \mathbb{R}^k$ and $G \subset \mathbb{R}^m$:

$$Prob\left(\widehat{\boldsymbol{U}} \in F, \widehat{\boldsymbol{B}} \in G\right) = \sum_{\boldsymbol{b} \in G} \int_F d\boldsymbol{u} \; \hat{\pi}(\boldsymbol{u}, \boldsymbol{b}).$$

We now define $\mathbf{Z} := \widehat{U} - A\widehat{B}$; since $\widehat{U}$ is continuous, $\mathbf{Z}$ is continuous too[1]. For any set $H \subset \mathbb{R}^k$ and $\mathbf{w} \in \mathbb{R}^k$, we denote by $H_{\mathbf{w}} := \{\mathbf{x} : \mathbf{x} - \mathbf{w} \in H\}$. We have that

$$
\begin{aligned}
Prob\left(\mathbf{Z} \in F, \ \widehat{B} \in G\right) &= Prob\left(\widehat{U} - A\widehat{B} \in F, \ \widehat{B} \in G\right) \\
&= \sum_{b \in G} Prob\left(\widehat{U} - A\widehat{B} \in F, \ \widehat{B} = b\right) \\
&= \sum_{b \in G} Prob\left(\widehat{U} \in F_{Ab}, \ \widehat{B} = b\right) \\
&= \sum_{b \in G} \int_{F_{Ab}} d\boldsymbol{u} \, \widehat{\pi}(\boldsymbol{u}, \boldsymbol{b}) \\
&= \sum_{b \in G} \int_{F} d\mathbf{z} \, \widehat{\pi}(\mathbf{z} + A\boldsymbol{b}, \boldsymbol{b}),
\end{aligned}
$$

where we used the change of variables $\mathbf{z} = \boldsymbol{u} - A\boldsymbol{b}$ in the integral. Hence, the density of $\left(\mathbf{Z}, \widehat{B}\right)$ with respect to $\mathcal{C}^m \otimes \mathcal{L}^k$ is:

$$
\pi_{(\mathbf{Z}, \widehat{B})}(\mathbf{z}, \boldsymbol{b}) := \widehat{\pi}(\mathbf{z} + A\boldsymbol{b}, \boldsymbol{b}). \tag{16}
$$

Note that the event $\left\{\widehat{Y} \in \mathcal{S}\right\}$ coincides with $\{\mathbf{Z} = \mathbf{0}\}$. As in Zambon et al. [2024b], we derive the expression of the reconciled distribution as the conditional density of $\widehat{B}$ given $\mathbf{Z} = \mathbf{0}$ [Çinlar, 2011, Chapter 4]:

$$
\begin{aligned}
\widetilde{\pi}(\boldsymbol{b}) &= \frac{\pi_{(\mathbf{Z}, \widehat{B})}(\mathbf{0}, \boldsymbol{b})}{\sum_{\mathbf{x}} \pi_{(\mathbf{Z}, \widehat{B})}(\mathbf{0}, \mathbf{x})} \\
&= \frac{\widehat{\pi}(A\boldsymbol{b}, \boldsymbol{b})}{\sum_{\mathbf{x}} \widehat{\pi}(A\mathbf{x}, \mathbf{x})} \\
&\propto \widehat{\pi}(A\boldsymbol{b}, \boldsymbol{b}),
\end{aligned}
$$

provided that $\sum_{\mathbf{x}} \widehat{\pi}(A\mathbf{x}, \mathbf{x}) > 0$.

---

[1]Let $F \subset \mathbb{R}^k$ be a measurable set such that $\mathcal{L}(F) = 0$. Then

$$
\begin{aligned}
Prob\left(\mathbf{Z} \in F\right) &= Prob\left(\widehat{U} - A\widehat{B} \in F\right) \\
&= \sum_{b \in \mathbb{N}} Prob\left(\widehat{B} = b, \widehat{U} \in F_{Ab}\right) \\
&\leq \sum_{b \in \mathbb{N}} Prob\left(\widehat{U} \in F_{Ab}\right) = 0,
\end{aligned}
$$

as $\mathcal{L}(F_{Ab}) = 0$ for any $\boldsymbol{b}$ (the Lebesgue measure is invariant under translations) and $\widehat{U}$ is continuous.

# B TOP-DOWN

In this section, for better readability, we sometimes use the integral notation even if the distributions are not continuous but discrete: e.g., we write $\int \pi(x,y)\,dy$ instead of $\sum_y \pi(x,y)$.

## B.1 BALANCED HIERARCHIES

Following Di Fonzo and Girolimetto [2024], we say that a hierarchy is *balanced* if each level of the hierarchy is complete. For example, the hierarchy in Fig. 9a is not balanced: the time series $u_1$ is equal to the sum of $u_2$, $u_3$, and $b_5$, and therefore the intermediate level is not complete. Note that any unbalanced hierarchy can be made balanced by duplicating some nodes. In this example, we can obtain a balanced hierarchy by adding the node $u_4$, which is just a copy of $b_5$ (Fig. 9b).

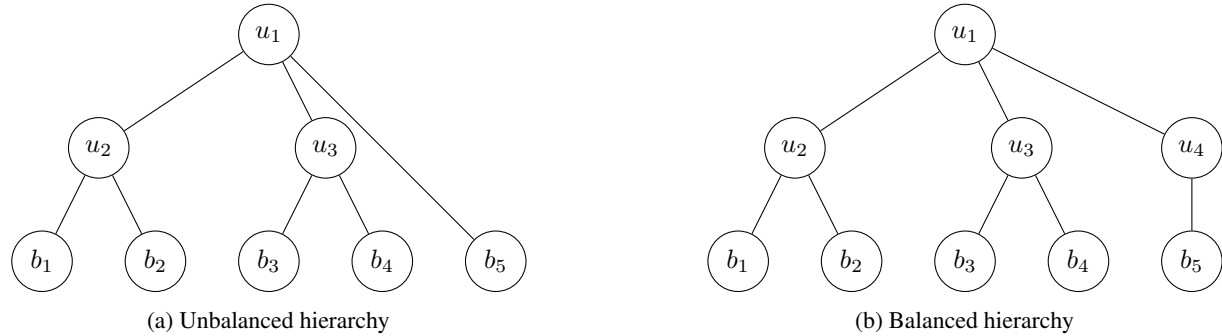

(a) Unbalanced hierarchy                    (b) Balanced hierarchy

Figure 9: The unbalanced hierarchy (left) is made balanced by duplicating the node $b_5$ (right)

If a hierarchy is balanced, there exists a set of "lowest upper time series" $\boldsymbol{u}^{low}$, such that any other upper time series is the sum of some lowest upper, and each bottom is child of only one of the lowest upper. For example, it is easy to see that such a set does not exist for the hierarchy of Fig. 9a, while $\boldsymbol{u}^{low} = [u_2, u_3, u_4]$ for the hierarchy of Fig. 9b. For any balanced hierarchy, we can consider the sub-hierarchy given by only the upper time series: the set of bottom time series of this sub-hierarchy is given by $\boldsymbol{u}^{low}$, so that there exists a matrix $\mathbf{S}^\mathbf{u}$ such that $\boldsymbol{u} = \mathbf{S}^\mathbf{u} \boldsymbol{u}^{low}$.

The assumption that the hierarchy is balanced is required by the *top-down conditioning* reconciliation approach. Indeed, it is needed for the first step of the algorithm, where the upper forecasts are reconciled via conditioning (lines 3-4 of Alg. 4); note that any hierarchy with only one upper time series is trivially balanced.

All the hierarchies used in the experiments in Sect. 5 are balanced. However, this is not a strong requirement, since any hierarchy can be made balanced by duplicating some bottom time series (Fig. 9).

## B.2 PROOF THAT $\widetilde{\pi}_{\mathrm{TD}}$ IS A PROBABILITY DISTRIBUTION

First, it is trivial from Eq. (6) that $\widetilde{\pi}_{\mathrm{TD}}(\boldsymbol{b}) \geq 0$ for any $\boldsymbol{b}$. Moreover, from Eq.(7) follows that, for any $\boldsymbol{b}$ such that $\widehat{\pi}_{bu}(b_1 + \cdots + b_m) = 0$, we have $\widehat{\pi}_1(b_1)\ldots\widehat{\pi}_m(b_m) = 0$; hence, for any $\boldsymbol{b}$ such that the denominator is equal to 0, the numerator is also 0. We implicitly define $\widetilde{\pi}_{\mathrm{TD}}$ as 0 on such $\boldsymbol{b}$. Finally, $\widetilde{\pi}_{\mathrm{TD}}$ is normalized:

$$\sum_{b_1,\ldots,b_m} \widetilde{\pi}_{\mathrm{TD}}(b_1,\ldots,b_m) = \sum_{b_1,\ldots,b_m} \widehat{\pi}_1(b_1)\ldots\widehat{\pi}_m(b_m) \frac{\widehat{\pi}_U(b_1 + \cdots + b_m)}{\widehat{\pi}_{bu}(b_1 + \cdots + b_m)}$$

$$= \sum_u \sum_{\substack{b_1,\ldots,b_m: \\ b_1+\cdots+b_m=u}} \widehat{\pi}_1(b_1)\ldots\widehat{\pi}_m(b_m) \frac{\widehat{\pi}_U(b_1 + \cdots + b_m)}{\widehat{\pi}_{bu}(b_1 + \cdots + b_m)}$$

$$= \sum_u \frac{\widehat{\pi}_U(u)}{\widehat{\pi}_{bu}(u)} \sum_{\substack{b_1,\ldots,b_m: \\ b_1+\cdots+b_m=u}} \widehat{\pi}_1(b_1)\ldots\widehat{\pi}_m(b_m)$$

$$= \sum_u \frac{\widehat{\pi}_U(u)}{\widehat{\pi}_{bu}(u)} \widehat{\pi}_{bu}(u) = \sum_u \widehat{\pi}_U(u) = 1.$$

## B.3  PROOF OF PROPOSITION 2

(1) Let $(\widetilde{B}_1, \ldots, \widetilde{B}_m) \sim \widetilde{\pi}_{\mathrm{TD}}$, and $\widetilde{U} := \widetilde{B}_1 + \cdots + \widetilde{B}_m$. Then, the distribution of $\widetilde{U}$ is given by

$$
\begin{aligned}
\pi_{\widetilde{U}}(u) &= \sum_{\substack{b_1,\ldots,b_m: \\ b_1+\cdots+b_m=u}} \widetilde{\pi}_{\mathrm{TD}}(b_1, \ldots, b_m) \\
&= \sum_{\substack{b_1,\ldots,b_m: \\ b_1+\cdots+b_m=u}} \frac{\widehat{\pi}_1(b_1) \ldots \widehat{\pi}_m(b_m)}{\widehat{\pi}_{bu}(b_1 + \cdots + b_m)} \widehat{\pi}_U(b_1 + \cdots + b_m) \\
&= \sum_{\substack{b_1,\ldots,b_m: \\ b_1+\cdots+b_m=u}} \frac{\widehat{\pi}_1(b_1) \ldots \widehat{\pi}_m(b_m)}{\widehat{\pi}_{bu}(u)} \widehat{\pi}_U(u) \\
&= \frac{\widehat{\pi}_U(u)}{\widehat{\pi}_{bu}(u)} \sum_{\substack{b_1,\ldots,b_m: \\ b_1+\cdots+b_m=u}} \widehat{\pi}_1(b_1) \ldots \widehat{\pi}_m(b_m) \\
&= \widehat{\pi}_U(u).
\end{aligned}
$$

Note that this holds only for $u$ belonging to the support of the bottom-up distribution, i.e. $u$ such that $\widehat{\pi}_{bu}(u) \neq 0$, as remarked in Sec. 4.

(2) Let $\bar{b}_1 + \cdots + \bar{b}_m = u = \check{b}_1 + \cdots + \check{b}_m$. Then

$$
\begin{aligned}
\frac{\widetilde{\pi}_{\mathrm{TD}}(\bar{b}_1, \ldots, \bar{b}_m)}{\widetilde{\pi}_{\mathrm{TD}}(\check{b}_1, \ldots, \check{b}_m)} &= \frac{\widehat{\pi}_1(\bar{b}_1) \ldots \widehat{\pi}_m(\bar{b}_m) \widehat{\pi}_U(u)}{\widehat{\pi}_{bu}(u)} \cdot \frac{\widehat{\pi}_{bu}(u)}{\widehat{\pi}_1(\check{b}_1) \ldots \widehat{\pi}_m(\check{b}_m) \widehat{\pi}_U(u)} \\
&= \frac{\widehat{\pi}_1(\bar{b}_1) \ldots \widehat{\pi}_m(\bar{b}_m)}{\widehat{\pi}_1(\check{b}_1) \ldots \widehat{\pi}_m(\check{b}_m)}.
\end{aligned}
$$

## B.4  PROOF OF LEMMA 1

The distribution of the output $(b_1, b_2)$ of Alg. 1 is given by

$$
\pi(b_1 \mid u) = \frac{\widehat{\pi}_1(b_1) \widehat{\pi}_2(u - b_1)}{\sum_b \widehat{\pi}_1(b) \widehat{\pi}_2(u - b)},
$$

$$
\pi(b_2 \mid b_1, u) = \mathbb{1}_{b_2 = u - b_1}.
$$

Hence

$$
\begin{aligned}
\pi(b_1, b_2 \mid u) &= \pi(b_2 \mid b_1, u) \, \pi(b_1 \mid u) \\
&= \frac{\widehat{\pi}_1(b_1) \widehat{\pi}_2(u - b_1)}{\sum_b \widehat{\pi}_1(b) \widehat{\pi}_2(u - b)} \mathbb{1}_{b_2 = u - b_1} \\
&= \frac{\widehat{\pi}_1(b_1) \widehat{\pi}_2(b_2)}{\sum_b \widehat{\pi}_1(b) \widehat{\pi}_2(u - b)} \mathbb{1}_{u = b_1 + b_2} \\
&= \widehat{\pi}(b_1, b_2 \mid b_1 + b_2 = u).
\end{aligned}
$$

## B.5  PROOF OF LEMMA 2

For each $l = 1, \ldots, L - 1$ and $j = 1, \ldots, 2^{L-l-1}$, consider $b^{(l)}_{2j-1}, b^{(l)}_{2j}$ from line 13 of Alg. 2. From Lemma 1, we have

$$
\begin{aligned}
\pi\left(b^{(l)}_{2j-1}, b^{(l)}_{2j} \mid b^{(l+1)}_j\right) &= \frac{\pi^{(l)}_{2j-1}\left(b^{(l)}_{2j-1}\right) \pi^{(l)}_{2j}\left(b^{(l)}_{2j}\right)}{\sum_b \pi^{(l)}_{2j-1}(b) \pi^{(l)}_{2j}\left(b^{(l+1)}_j - b\right)} \mathbb{1}_{b^{(l+1)}_j = b^{(l)}_{2j-1} + b^{(l)}_{2j}} \\
&= \frac{\pi^{(l)}_{2j-1}\left(b^{(l)}_{2j-1}\right) \pi^{(l)}_{2j}\left(b^{(l)}_{2j}\right)}{\pi^{(l+1)}_j\left(b^{(l+1)}_j\right)} \mathbb{1}_{b^{(l+1)}_j = b^{(l)}_{2j-1} + b^{(l)}_{2j}},
\end{aligned} \tag{17}
$$

where the denominator in second equation is the result of the convolution in line 8 of Alg. 2. If we denote by $\boldsymbol{b}^{(l)} = \left(b_1^{(l)}, \ldots, b_{2^{L-l}}^{(l)}\right)$, for each $l = 1, \ldots, L$, we obtain

$$
\begin{aligned}
\pi\left(\boldsymbol{b}^{(l)} \mid \boldsymbol{b}^{(l+1)}\right) &= \prod_{j=1}^{2^{L-l-1}} \pi\left(b_{2j-1}^{(l)}, b_{2j}^{(l)} \mid b_j^{(l+1)}\right) \\
&= \frac{\prod_{j=1}^{2^{L-l}} \pi_j^{(l)}\left(b_j^{(l)}\right)}{\prod_{j=1}^{2^{L-l-1}} \pi_j^{(l+1)}\left(b_j^{(l+1)}\right)} \prod_{j=1}^{2^{L-l-1}} \mathbb{1}_{b_j^{(l+1)} = b_{2j-1}^{(l)} + b_{2j}^{(l)}},
\end{aligned}
\tag{18}
$$

where the second equation is the result of plugging-in Eq. (17). Hence

$$
\begin{aligned}
\pi\left(\boldsymbol{b}^{(1)}, \boldsymbol{b}^{(2)}, \ldots, \boldsymbol{b}^{(L)} \mid u\right) &= \prod_{l=1}^{L-1} \pi\left(\boldsymbol{b}^{(l)} \mid \boldsymbol{b}^{(l+1)}\right) \pi\left(\boldsymbol{b}^{(L)} \mid u\right) \\
&= \prod_{l=1}^{L-1} \frac{\prod_{j=1}^{2^{L-l}} \pi_j^{(l)}\left(b_j^{(l)}\right)}{\prod_{j=1}^{2^{L-l-1}} \pi_j^{(l+1)}\left(b_j^{(l+1)}\right)} \prod_{j=1}^{2^{L-l-1}} \mathbb{1}_{b_j^{(l+1)} = b_{2j-1}^{(l)} + b_{2j}^{(l)}} \cdot \mathbb{1}_{b_1^{(L)} = u} \\
&= \frac{\prod_{j=1}^{m} \pi_j^{(1)}\left(b_j^{(1)}\right)}{\pi_1^{(L)}\left(b_1^{(L)}\right)} \prod_{l=1}^{L-1} \prod_{j=1}^{2^{L-l-1}} \mathbb{1}_{b_j^{(l+1)} = b_{2j-1}^{(l)} + b_{2j}^{(l)}} \cdot \mathbb{1}_{b_1^{(L)} = u},
\end{aligned}
\tag{19}
$$

where we use Eq. (18) and line 10 of Alg. 2 in the second equation, and the third equation is the result of a telescoping product. Therefore

$$
\begin{aligned}
\pi\left(b_1^{(1)}, \ldots, b_m^{(1)} \mid u\right) &= \pi\left(\boldsymbol{b}^{(1)} \mid u\right) \\
&= \int \pi\left(\boldsymbol{b}^{(1)}, \boldsymbol{b}^{(2)}, \ldots, \boldsymbol{b}^{(L)} \mid u\right) d\boldsymbol{b}^{(2)} \ldots d\boldsymbol{b}^{(L)} \\
&= \int \frac{\prod_{j=1}^{m} \pi_j^{(1)}\left(b_j^{(1)}\right)}{\pi_1^{(L)}\left(b_1^{(L)}\right)} \prod_{l=1}^{L-1} \prod_{j=1}^{2^{L-l-1}} \mathbb{1}_{b_j^{(l+1)} = b_{2j-1}^{(l)} + b_{2j}^{(l)}} \cdot \mathbb{1}_{b_1^{(L)} = u} \, d\boldsymbol{b}^{(2)} \ldots d\boldsymbol{b}^{(L)} \\
&= \frac{\prod_{j=1}^{m} \pi_j^{(1)}\left(b_j^{(1)}\right)}{\pi_1^{(L)}(u)} \mathbb{1}_{b_1^{(1)} + \cdots + b_m^{(1)} = u}.
\end{aligned}
\tag{20}
$$

Since $\pi_j^{(1)} = \widehat{\pi}_j$, for all $j = 1, \ldots, m$, and $\pi_1^{(L)} = \pi_1^{(1)} * \cdots * \pi_m^{(1)} = \widehat{\pi}_{bu}$, we conclude from Eq. (20) that

$$
\begin{aligned}
\pi\left(b_1^{(1)}, \ldots, b_m^{(1)} \mid u\right) &= \frac{\prod_{j=1}^{m} \widehat{\pi}_j\left(b_j^{(1)}\right)}{\widehat{\pi}_{bu}(u)} \mathbb{1}_{b_1^{(1)} + \cdots + b_m^{(1)} = u} \\
&= \widehat{\pi}\left(b_1^{(1)}, \ldots, b_m^{(1)} \mid b_1^{(1)} + \cdots + b_m^{(1)} = u\right).
\end{aligned}
$$

## B.6   PROOF OF PROPOSITION 3

From Lemma 2 follows that, for all $i = 1, \ldots, N$:

$$
\pi\left(\boldsymbol{b}^i \mid u^i\right) = \frac{\widehat{\pi}_1\left(b_1^i\right) \ldots \widehat{\pi}_m\left(b_m^i\right)}{\widehat{\pi}_{bu}\left(u^i\right)} \mathbb{1}_{b_1^i + \cdots + b_m^i = u^i}.
$$

Since $u^i \sim \widehat{\pi}_U$, we have

$$
\begin{aligned}
\pi(\boldsymbol{b}^i) &= \int \pi(\boldsymbol{b}^i, u^i) \, du^i \\
&= \int \pi(\boldsymbol{b}^i \mid u^i) \, \pi(u^i) \, du^i \\
&= \int \frac{\widehat{\pi}_1(b_1^i) \dots \widehat{\pi}_m(b_m^i)}{\widehat{\pi}_{bu}(u^i)} \, \mathbb{1}_{b_1^i + \dots + b_m^i = u^i} \, \pi(u^i) \, du^i \\
&= \frac{\widehat{\pi}_1(b_1^i) \dots \widehat{\pi}_m(b_m^i)}{\widehat{\pi}_{bu}(b_1^i + \dots + b_m^i)} \, \pi(b_1^i + \dots + b_m^i) \\
&= \widetilde{\pi}_{\mathrm{TD}}(\boldsymbol{b}^i).
\end{aligned}
$$

## B.7 EXTENSION OF ALGORITHM 2 TO GENERIC $m$

We now consider the case of a hierarchy with one upper and $m$ bottom time series, but we drop the assumption that $m$ is a power of 2. In this case, we proceed as follows. Let $M$ be the smallest power of 2 greater or equal to $m$, i.e. $M = 2^{\lceil \log_2(m) \rceil}$, and let $L := \log_2(M) + 1$. We build a binary tree with $L$ levels as in Sec. 4.1. In this case however the "missing" bottom nodes have distribution given by a Dirac's delta centered in 0, denoted by $\delta_0$:

$$
\begin{aligned}
B_j^{(1)} &= \widehat{B}_j, & &\text{for } j = 1, \dots, m, \\
B_j^{(1)} &= \delta_0, & &\text{for } j = m+1, \dots, M, \\
B_j^{(l+1)} &= B_{2j-1}^{(l)} + B_{2j}^{(l)}, & &\text{for } l = 1, \dots, L-1, \ j = 1, \dots, 2^{L-l-1}.
\end{aligned}
$$

An example for $m = 3$ is shown in Fig. 10. As in Sec. 4.1, we also denote by $\pi_j^{(l)}$ the distribution of $B_j^{(l)}$, so that

$$
\begin{aligned}
\pi_j^{(1)}(b_j) &= \widehat{\pi}_j(b_j), & &\text{for } j = 1, \dots, m, \\
\pi_j^{(1)}(b_j) &= \mathbb{1}_{b_j = 0}, & &\text{for } j = m+1, \dots, M, \\
\pi_j^{(l+1)} &= \pi_{2j-1}^{(l)} * \pi_{2j}^{(l)}, & &\text{for } l = 1, \dots, L-1, \ j = 1, \dots, 2^{L-l-1}.
\end{aligned}
$$

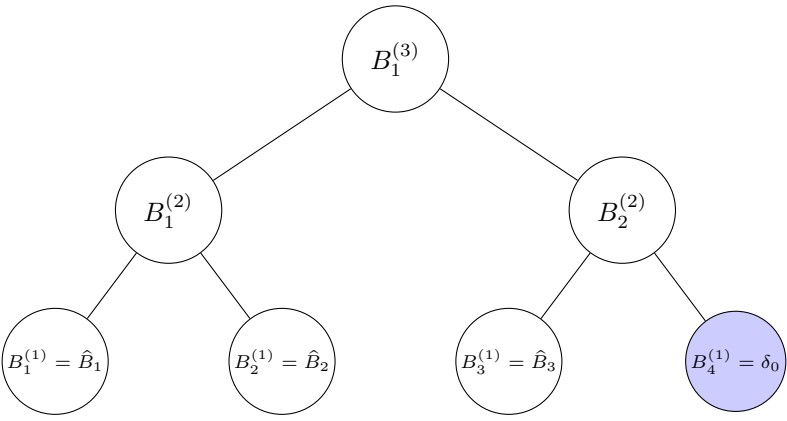

Figure 10: Binary tree ($m = 3$, $L = 3$)

We can then run Alg. 2 with $\pi_1^{(1)}, \dots, \pi_M^{(1)}$, and only keep the first $m$ terms of the sample $(b_1, \dots, b_m, b_{m+1}, \dots, b_M)$. Indeed, from Lemma 2

$$
\begin{aligned}
\pi(b_1, \dots, b_M \mid u) &= \frac{\pi_1^{(1)}(b_1) \dots \pi_M^{(1)}(b_M)}{\widehat{\pi}_{bu(1:M)}(u)} \, \mathbb{1}_{u = b_1 + \dots + b_M} \\
&= \frac{\widehat{\pi}_1(b_1) \dots \widehat{\pi}_m \, \mathbb{1}_{b_{m+1} = 0} \dots \mathbb{1}_{b_M = 0}}{\widehat{\pi}_{bu(1:M)}(u)} \, \mathbb{1}_{u = b_1 + \dots + b_M},
\end{aligned}
$$

where

$$\widehat{\pi}_{bu(1:M)} = \pi_1^{(1)} * \cdots * \pi_M^{(1)}$$
$$= \widehat{\pi}_1 * \cdots * \widehat{\pi}_m * \delta_0 * \cdots * \delta_0$$
$$= \widehat{\pi}_1 * \cdots * \widehat{\pi}_m = \widehat{\pi}_{bu(1:m)}.$$

By integrating out $b_{m+1}, \ldots, b_M$, we obtain

$$\pi(b_1, \ldots, b_m \,|\, u) = \int \pi(b_1, \ldots, b_M \,|\, u)\, db_{m+1} \ldots db_M$$
$$= \int \frac{\widehat{\pi}_1(b_1) \ldots \widehat{\pi}_m(b_m)\, \mathbb{1}_{b_{m+1}=0} \ldots \mathbb{1}_{b_M=0}}{\widehat{\pi}_{bu(1:m)}(u)}\, \mathbb{1}_{u=b_1+\cdots+b_M}\, db_{m+1} \ldots db_M$$
$$= \frac{\widehat{\pi}_1(b_1) \ldots \widehat{\pi}_m(b_m)}{\widehat{\pi}_{bu(1:m)}(u)}\, \mathbb{1}_{u=b_1+\cdots+b_m}$$

**Remark 1.** *In practice, since $\pi * \delta_0 = \pi$ for any distribution $\pi$, when we run the algorithm there is no need to compute any of the convolutions with the "missing" nodes. For example, for the binary tree of Fig. 10, when we reach $B_2^{(2)}$ we can stop as we have $B_3^{(1)} = B_2^{(2)}$.*

## B.8  PROOF OF PROPOSITION 4

(1) Let $\widetilde{B} \sim \widetilde{\pi}_{\mathrm{TD}}$, and $\widetilde{U} := A\widetilde{B}$. Then, the distribution of $\widetilde{U}$ is given by

$$\pi_{\widetilde{U}}(\boldsymbol{u}) = \sum_{\boldsymbol{b}:\, \boldsymbol{A}\boldsymbol{b}=\boldsymbol{u}} \widetilde{\pi}_{\mathrm{TD}}(\boldsymbol{b})$$
$$= \sum_{\boldsymbol{b}:\, \boldsymbol{A}\boldsymbol{b}=\boldsymbol{u}} \frac{\widehat{\pi}_1(b_1) \ldots \widehat{\pi}_m(b_m)}{\widehat{\pi}_{bu}(\boldsymbol{A}\boldsymbol{b})}\, \widehat{\pi}_U(\boldsymbol{A}\boldsymbol{b})$$
$$= \sum_{\boldsymbol{b}:\, \boldsymbol{A}\boldsymbol{b}=\boldsymbol{u}} \frac{\widehat{\pi}_1(b_1) \ldots \widehat{\pi}_m(b_m)}{\widehat{\pi}_{bu}(\boldsymbol{u})}\, \widehat{\pi}_U(\boldsymbol{u})$$
$$= \mathbb{1}_{\boldsymbol{u}^{upp}=\mathbf{A}^{\mathbf{u}}\boldsymbol{u}^{low}}\, \frac{\widehat{\pi}_U(\boldsymbol{u})}{\widehat{\pi}_{bu}(\boldsymbol{u})} \sum_{\boldsymbol{b}:\, \boldsymbol{A}\boldsymbol{b}=\boldsymbol{u}} \widehat{\pi}_1(b_1) \ldots \widehat{\pi}_m(b_m)$$
$$= \mathbb{1}_{\boldsymbol{u}^{upp}=\mathbf{A}^{\mathbf{u}}\boldsymbol{u}^{low}}\, \widehat{\pi}_U(\boldsymbol{u}).$$

Note that the indicator function is necessary, since $1/\widehat{\pi}_{bu}(\boldsymbol{u})$ can be pulled out of the sum only if $\widehat{\pi}_{bu}(\boldsymbol{u}) \neq 0$, which holds if $\boldsymbol{u}^{upp} = \mathbf{A}^{\mathbf{u}}\boldsymbol{u}^{low}$.

(2) Let $\boldsymbol{A}\bar{\mathbf{b}} = \boldsymbol{u} = \boldsymbol{A}\check{\mathbf{b}}$. Then

$$\frac{\widetilde{\pi}_{\mathrm{TD}}(\bar{\mathbf{b}})}{\widetilde{\pi}_{\mathrm{TD}}(\check{\mathbf{b}})} = \frac{\widehat{\pi}_1(\bar{b}_1) \ldots \widehat{\pi}_m(\bar{b}_m)\, \widehat{\pi}_U(\boldsymbol{u})}{\widehat{\pi}_{bu}(\boldsymbol{u})} \cdot \frac{\widehat{\pi}_{bu}(\boldsymbol{u})}{\widehat{\pi}_1(\check{b}_1) \ldots \widehat{\pi}_m(\check{b}_m)\, \widehat{\pi}_U(\boldsymbol{u})}$$
$$= \frac{\widehat{\pi}_1(\bar{b}_1) \ldots \widehat{\pi}_m(\bar{b}_m)}{\widehat{\pi}_1(\check{b}_1) \ldots \widehat{\pi}_m(\check{b}_m)}.$$

## B.9  PROOF OF PROPOSITION 5

We need to prove that

$$\left(\bar{\mathbf{b}}^{(1)}, \ldots, \bar{\mathbf{b}}^{(k_{low})}\right) \sim \frac{\widehat{\pi}_1(b_1) \ldots \widehat{\pi}_m(b_m)}{\widehat{\pi}_{bu}(\boldsymbol{A}\boldsymbol{b})}\, \widehat{\pi}_U(\boldsymbol{A}\boldsymbol{b}). \tag{21}$$

We denote $\bar{\mathbf{b}}^{(j)} = \left(\bar{\mathbf{b}}_1^{(j)}, \ldots, \bar{\mathbf{b}}_{m_j}^{(j)}\right)$, for all $j = 1, \ldots, k_{low}$, and we drop the superscript $i$ for better readability. First, from line 8 of Alg. 4, and from Lemma 2, we have that

$$\bar{\mathbf{b}}^{(j)} \,|\, u_j \sim \frac{\widehat{\pi}(\bar{\mathbf{b}}_1^{(j)}) \ldots \widehat{\pi}(\bar{\mathbf{b}}_{m_j}^{(j)})}{\widehat{\pi}_{bu}(u_j)}\, \mathbb{1}_{u_j=\bar{\mathbf{b}}_1^{(j)}+\cdots+\bar{\mathbf{b}}_{m_j}^{(j)}},$$

and therefore

$$\pi\left(\bar{\mathbf{b}}^{(1)},\ldots,\bar{\mathbf{b}}^{(k_{low})}\,\big|\,u_1,\ldots,u_{k_{low}}\right) = \frac{\widehat{\pi}_1(b_1)\ldots\widehat{\pi}_m(b_m)}{\widehat{\pi}_{bu}(u_1)\ldots\widehat{\pi}_{bu}(u_{k_{low}})}\,\mathbb{1}_{u_1=\bar{\mathbf{b}}_1^{(1)}+\cdots+\bar{\mathbf{b}}_{m_1}^{(1)}}\cdots\mathbb{1}_{u_{k_{low}}=\bar{\mathbf{b}}_1^{(k_{low})}+\cdots+\bar{\mathbf{b}}_{m_{k_{low}}}^{(k_{low})}}. \quad (22)$$

Moreover, from line 4 of Alg. 4 and Eq. (4):

$$\pi(u_1,\ldots,u_{k_{low}}) \propto \widehat{\pi}_U\left(\mathbf{A^u}\boldsymbol{u}^{low},\,\boldsymbol{u}^{low}\right), \quad (23)$$

where $\boldsymbol{u}^{low} = (u_1,\ldots,u_{k_{low}})$. Joining Eq. (22) and Eq. (23), and integrating out $u_1,\ldots,u_{k_{low}}$, we obtain

$$\pi\left(\bar{\mathbf{b}}^{(1)},\ldots,\bar{\mathbf{b}}^{(k_{low})}\right) = \int \pi\left(\bar{\mathbf{b}}^{(1)},\ldots,\bar{\mathbf{b}}^{(k_{low})}\,\big|\,u_1,\ldots,u_{k_{low}}\right)\pi(u_1,\ldots,u_{k_{low}})\,d\boldsymbol{u}^{low}$$

$$\propto \widehat{\pi}_1(b_1)\ldots\widehat{\pi}_m(b_m)\int \frac{\widehat{\pi}_U\left(\mathbf{A^u}\boldsymbol{u}^{low},\,\boldsymbol{u}^{low}\right)\mathbb{1}_{u_1=\bar{\mathbf{b}}_1^{(1)}+\cdots+\bar{\mathbf{b}}_{m_1}^{(1)}}\cdots\mathbb{1}_{u_{k_{low}}=\bar{\mathbf{b}}_1^{(k_{low})}+\cdots+\bar{\mathbf{b}}_{m_{k_{low}}}^{(k_{low})}}}{\widehat{\pi}_{bu}(u_1)\ldots\widehat{\pi}_{bu}(u_{k_{low}})}\,d\boldsymbol{u}^{low}$$

$$= \widehat{\pi}_1(b_1)\ldots\widehat{\pi}_m(b_m)\int \frac{\widehat{\pi}_U\left(\mathbf{A^u}\boldsymbol{u}^{low},\,\boldsymbol{u}^{low}\right)\mathbb{1}_{u_1=\bar{\mathbf{b}}_1^{(1)}+\cdots+\bar{\mathbf{b}}_{m_1}^{(1)}}\cdots\mathbb{1}_{u_{k_{low}}=\bar{\mathbf{b}}_1^{(k_{low})}+\cdots+\bar{\mathbf{b}}_{m_{k_{low}}}^{(k_{low})}}}{\displaystyle\sum_{\substack{\boldsymbol{b}^{(j)}:\,b_1^{(j)}+\cdots+b_{m_j}^{(j)}=u_j\\ j=1,\ldots,k_{low}}}\widehat{\pi}(b_1^{(1)})\ldots\widehat{\pi}(b_{m_1}^{(1)})\ldots\widehat{\pi}(b_1^{(k_{low})})\ldots\widehat{\pi}(b_{m_{k_{low}}}^{(k_{low})})}\,d\boldsymbol{u}^{low}$$

$$\propto \frac{\widehat{\pi}_1(b_1)\ldots\widehat{\pi}_m(b_m)}{\widehat{\pi}_{bu}(\boldsymbol{Ab})}\,\widehat{\pi}_U(\boldsymbol{Ab}).$$