# OpenReview forum: "Probabilistic reconciliation of mixed-type hierarchical time series"
_auai.org/UAI/2024/Conference — UAI 2024 poster_

### Official Review · Reviewer_Gma3 · 2024-03-16

**Q2-1 Originality-Novelty:** 3
**Q2-2 Correctness-Technical Quality:** 3
**Q2-5 Clarity Of Writing:** 3

**Q1 Summary And Contributions:**

The authors propose two approaches for probabilistic reconciliation of hierarchical forecasts under the mixed setting with continuous upper and discrete lower. Some theoretical properties of the proposed approach are presented. Empirical results on three datasets are presented where the proposed approaches are shown to outperform existing baselines.

**Q2-3 Extent To Which Claims Are Supported By Evidence:**

3: Good: the main claims are supported by convincing evidence (in the form of adequate experimental evaluation, proofs, (pseudo-)code, references, assumptions).

**Q2-4 Reproducibility:**

2: Fair: key resources (e.g. proofs, code, data) are unavailable but key details (e.g. proof sketches, experimental setup) are sufficiently well-described for an expert to confidently reproduce the main results.

**Q3 Main Strengths:**

The paper addresses an existing open problem in probabilistic reconciliation between predictions of mixed type variables within a hierarchy. Although this appears to be a narrow area with limited target audience, the work still makes non-trivial contribution to an important problem with practical impact. The empirical evaluation is adequate and justifies the need for the proposed approaches.

**Q4 Main Weakness:**

The independence assumption is made throughout for all the results, which is less than ideal since in practice this is hardly the case. For a sampling-based approach, I wonder whether a more relaxed assumption is readily achievable.

**Q5 Detailed Comments To The Authors:**

See above.

**Q9 Complying With Reviewing Instructions:**

Yes

---

> ### Author Rebuttal · Authors · 2024-04-04
>
> We thank the reviewer for his comments.
>
> Indeed, the independence assumption is often unrealistic. However, there are currently no methods available in the literature to model correlations between discrete univariate forecast distributions. The sampling algorithm, in its current form, also requires the hypothesis of independence between the base forecasts: relaxing this assumption is therefore not readily achievable. This is an interesting research direction that we leave for future work. We will add this point to the conclusions of the paper.

---

### Official Review · Reviewer_CkzD · 2024-03-19

**Q2-1 Originality-Novelty:** 2
**Q2-2 Correctness-Technical Quality:** 3
**Q2-5 Clarity Of Writing:** 3

**Q1 Summary And Contributions:**

In this paper the authors presented two methods for reconciliation of incoherent forecasting for hierarchical structure time series with mixed distributions. The authors then applied the methods to several data sets to evaluate the strength and weakness of the two methods.

**Q2-3 Extent To Which Claims Are Supported By Evidence:**

3: Good: the main claims are supported by convincing evidence (in the form of adequate experimental evaluation, proofs, (pseudo-)code, references, assumptions).

**Q2-4 Reproducibility:**

3: Good: key resources (e.g. proofs, code, data) are available and key details (e.g. proofs, experimental setup) are sufficiently well-described for competent researchers to confidently reproduce the main results.

**Q3 Main Strengths:**

1. The authors presented two methods for reconciliation of incoherent forecasting for hierarchical structure time series with mixed distributions that are suitable for small and large data set.
2. The authors showed the methods outperform the current continuous methods, especially for bottom time series

**Q4 Main Weakness:**

1. The contribution is incremental.
2. The continuous methods seem doing fine for upper time series, and outperform the Top-down method for small data.

**Q5 Detailed Comments To The Authors:**

There is an extra "dx" in the last line of page 12.

**Q9 Complying With Reviewing Instructions:**

Yes

---

> ### Author Rebuttal · Authors · 2024-04-04
>
> We thank the reviewer for his comments; below are our answers to the comments in Q4.
>
> 1. This is the first paper that reconciles mixed hierarchies. This is explicitly mentioned as an important open problem in the recent and authoritative review on forecast reconciliation (Athanasopoulos, Hyndman, Kourentzes, Panagiotelis, “Forecast reconciliation: A Review”, Int. J. Forecast., 2024). No methods are currently available to tackle this problem, which is common in many applied scenarios.
>
> 2. The Gaussian approach gives good results only on small datasets, and only on the upper levels: this is why we need better-suited methods in the case of mixed hierarchies.

---

### Official Review · Reviewer_2QNa · 2024-03-20

**Q2-1 Originality-Novelty:** 3
**Q2-2 Correctness-Technical Quality:** 3
**Q2-5 Clarity Of Writing:** 3

**Q1 Summary And Contributions:**

The paper presents two approaches to reconcile hierarchical time series that contain discrete and continuous time series: mixed conditioning and top-down conditioning. The former uses conditioning as defined by Zambon et al. The latter consists of two steps, first the time series in the higher levels of the hierarchy are reconciled using conditioning and then, the bottom time series are updated. A set of experiments highlights the two approaches.

**Q2-3 Extent To Which Claims Are Supported By Evidence:**

3: Good: the main claims are supported by convincing evidence (in the form of adequate experimental evaluation, proofs, (pseudo-)code, references, assumptions).

**Q2-4 Reproducibility:**

3: Good: key resources (e.g. proofs, code, data) are available and key details (e.g. proofs, experimental setup) are sufficiently well-described for competent researchers to confidently reproduce the main results.

**Q3 Main Strengths:**

- Clear motivation, problem setting, solution approaches
- Mostly complete, self-contained formalisation

**Q4 Main Weakness:**

- Presentation could be improved; a bit hard to read for someone who has worked with time series but never worked on reconciliation

**Q5 Detailed Comments To The Authors:**

The formalism is set up nicely but it would benefit if the notions of counts for the bottom time series and the predictive distributions for the upper time series (e.g., in Section 3; Section 4 contains "count time series") would be formally defined or at least anchored in the definitions set up in Section 2.

Minor comments
-------------
- I recognise that upper / bottom are used as a shorthand but they do not make nice, grammatically correct English; please consider replacing upper / bottom with longer versions because sometimes they appear to be a shorthand for time series, forecasts, distributions; most of the time, it is clear from context what it means in a sentence but it would make reading easier to have it explicitly written (and as mentioned it would make sentences grammatically correct; e.g., "where the bottom are defined" clashes on singular vs. plural between "bottom" (sg) and "are defined" (pl); "between the upper")
- A heading should usually be followed by a(n introductory) sentence; having Fig. 1 directly after the heading of Section 2 is a bit weird and confusing because one has to look for the start of the section and has no clue what to do with the figure at this point (an introductory sentence instead of starting directly with "Fig. 1 shows..." would also help guide the reader
- Fig., App., Prop. are all capitalised but eq. is not -> capitalise as well for consistency?
- Sec. 2, "obtained via eq.(2)" -> is there a space missing between eq. and (2)?
- Space between Prop. 1 and the following paragraph appears overly large -> superfluous line break / empty line?
- Figure 2 is not explicitly referenced in the paper if I am not mistaken.
- Please think of colour-blind people (and grey-scale print-outs) when setting up the figures; the red and green over each other will not be distinguishable
- Sec. 4, "A similar idea has been given by Elgavish" -> sounds overly complicated, active voice? "Elgavish presents a similar idea"?
- Sec. 4 misses a few Eq. in front of (x) referencing equations, which the other sections use -> consistency
- Sec. 4.1: "On the other hand" -> requires an "on the one hand" to have come before
- It would make for a better text if the properties in Prop. 2 and Prop. 4 would receive an explanation as to why they are useful
- Algorithm 1 uses b but b does not appear as an input; it would also be nice to get a bit more information about how "Define ..." in line 3 is supposed to work in the text
- Please explain in the text what the purpose of Lemma 1 is; the lemma appears a bit contextless in the text

**Q9 Complying With Reviewing Instructions:**

Yes

---

> ### Author Rebuttal · Authors · 2024-04-04
>
> We thank the reviewer for all the comments, which we will implement in the revised version of the paper. In particular, we will expand Sect. 2 to include a more detailed explanation of hierarchical forecasting, reconciliation, predictive distributions, and count time series.
>
> We also clarify that the purpose of Lemma 1 is to express what is the distribution of the output of Alg. 1, which is at the core of the more complex top-down sampling algorithm (Alg. 2).

---

### Official Review · Reviewer_gPaT · 2024-03-21

**Q2-1 Originality-Novelty:** 2
**Q2-2 Correctness-Technical Quality:** 2
**Q2-5 Clarity Of Writing:** 3

**Q10 Ethical Concerns:**

No.

**Q1 Summary And Contributions:**

The paper addresses the challenge of reconciling hierarchical time series forecasts, particularly focusing on mixed hierarchies where the there is a mixture of discrete and continuous time series. They propose two approaches: mixed conditioning and top-down conditioning. Mixed conditioning creates a joint distribution of base forecasts and applies coherence constraints, but it is not appropriate for large hierarchies. Top-down conditioning reconciles upper forecasts first, then updates bottom distributions. The latter approach handles large hierarchies.

**Q2-3 Extent To Which Claims Are Supported By Evidence:**

2: Fair: the main claims are somewhat supported by evidence (but the experimental evaluation may be weak, or does not match entirely with the claims, important baselines may be missing, proofs contain important ideas but lack rigor, algorithmic details are only discussed superficially, references are imprecise, assumptions are not sufficiently motivated or explicated, etc.).

**Q2-4 Reproducibility:**

3: Good: key resources (e.g. proofs, code, data) are available and key details (e.g. proofs, experimental setup) are sufficiently well-described for competent researchers to confidently reproduce the main results.

**Q3 Main Strengths:**

The proposed approach can handle both discrete and continuous time series.
It is a well-written paper, and they provided the proofs and pseudo algorithm for each scenario.
They also discuss the shortcoming of one of the proposed model, and the solution to address that.

**Q4 Main Weakness:**

They only applied the proposed methods on one data set, they did not assess the performance on simulated data, and under different settings, and assumptions.

**Q5 Detailed Comments To The Authors:**

1. Assessing the performance of the models on the simulated data can improve the study.
2. I am not sure if these models are generalizable to every data set, as it only applied only on one specific data set.
3. How we would know that the required assumptions are held in data set?
4. They considered the general case of a hierarchy with m bottom and k upper, single upper, does that make sense to consider the single bottom and k upper?
5. In case of high-dimensional dataset, they include many bottom time series but not many upper time series, what does the high dimension mean in this case? how the results would be changed if there are more upper time series or more upper and bottom time series at the same time?

**Q9 Complying With Reviewing Instructions:**

Yes

---

> ### Author Rebuttal · Authors · 2024-04-04
>
> We thank the reviewer for the comments and suggestions. Below are our detailed answers.
>
> 1. We only use real datasets of different sizes (syph-small, syph, and M5), as this is enough to highlight in which situation each proposed algorithm is preferable. Table 1 shows that syph and syph-small are moderately sized (less than 100 time series), while M5 is large (thousands of time series). The variety of datasets highlights under which conditions mix-cond is preferable to TD-cond and vice versa (the Gaussian methods are generally worse in each setting).
>
> 2. Our models are generalizable to any dataset: the only requirement is that the data structure is a proper hierarchy, i.e., it can be represented as a tree; this is often the case in hierarchical forecasting. We use three different datasets; in Table 2 we report the results for each dataset. Notice that M5, though appearing as a single data set in Table 2, contains 10 different datasets, one for each store (all with the same hierarchical structure). This is a large dataset that has never been probabilistically reconciled before.
>
>  3. We do not need any assumption on the dataset. We only need the hierarchy to be proper, and a probabilistic forecasting model to produce the base forecasts. We will make this point clearer in the paper.
>
> 4. Since the upper time series are aggregations of the bottom time series, we cannot have a single bottom and k upper time series.
>
> 5. Since the hierarchy is represented by a tree, the number of upper time series is always lower than the number of bottom time series. Anyway, the TD-cond method reconciles the upper part of the hierarchy analytically. This step only requires the inversion of a matrix and is therefore feasible with up to thousands of time series; this method is thus usable in most practical cases.

---

### Official Review · Reviewer_LKrg · 2024-03-25

**Q2-1 Originality-Novelty:** 3
**Q2-2 Correctness-Technical Quality:** 3
**Q2-5 Clarity Of Writing:** 3

**Q1 Summary And Contributions:**

This paper proposed two methods for probabilistic reconcilation for mixed-type hierarchical time series. Mixed-type hierarchical reconcilation is an important and less studied problem in time series forecasting. The two methods are mixed-conditioning and top down conditioning.

**Q2-3 Extent To Which Claims Are Supported By Evidence:**

3: Good: the main claims are supported by convincing evidence (in the form of adequate experimental evaluation, proofs, (pseudo-)code, references, assumptions).

**Q2-4 Reproducibility:**

2: Fair: key resources (e.g. proofs, code, data) are unavailable but key details (e.g. proof sketches, experimental setup) are sufficiently well-described for an expert to confidently reproduce the main results.

**Q3 Main Strengths:**

- This paper tackles an important and less studied problem with decent results.
- The method is explained in a logical manner. Experiments are well-designed, showing the the effectiveness of the proposed methods on the three datasets.

**Q4 Main Weakness:**

- The method can be explained better with a more detailed explanation on hiearchical forecasting. It is not clear what upper, bottom and base forecasts are. It is also not clear what the actual objective is, is it to improve accuracy or something else?
- The experiments are conducted well, however more datasets and methods should be tested. It is not clear how the method compares against normal forecasting method without doing hierarchical forecasting.

**Q5 Detailed Comments To The Authors:**

Overall, this is a nice paper that proposes two methods for hierarchical forecasting that are able to handle mixed distributions in the hierarchies.

I believe it would be good to give a more detailed explanations on hierarchical forecasting before explaining the method. For instance, it is not clear what upper, bottom and base forecasts are. More importantly what is the objective of reconcilation in hierarchial forecasting? Is it for improving accuracy or something else?

It would also be good to explain the figures in more details. I did not understand what the figures on the different distributions are trying to show.

The experiments are conducted well but more datasets and methods should be compared. It is not clear how the proposed methods compares to a non-hierarchical forecasting method.

It is hard to understand table 2. I have no idea what the values mean. It seems like negative scores mean poor performance, i.e. the proposed method is better than the baseline? If that is the case, what is the reason why the pprformance is poor on some datasets?

It seems like the methods can only handle two levels, upper and bottom. For example in M5, there are multiple levels and three levels are merged into upper level. Should a study be conducted for the different combinations? The levels are defined by humans and sometimes there could be intermediate levels in between that are not defined, such as monthly as bottom level, yearly as upper level and quarterly as the intermediate level. In such cases, how would one apply these methods?

**Q9 Complying With Reviewing Instructions:**

Yes

---

> ### Author Rebuttal · Authors · 2024-04-04
>
> We thank the reviewer for the comments and suggestions. Below are our detailed answers.
>
> **Reproducibility**. We provide the proofs in the Appendix; the code and data for reproducing our results are available in the supplementary material.
>
> **Hierarchical forecasting**. We provide here some background on hierarchical forecasting; we will include a similar explanation in the revised version of the paper. Hierarchical time series are collections of time series that are formed via aggregation and therefore satisfy some summing constraints. For instance, in Fig. 1, the time series u1 is equal to the sum of the time series u2 and u3, and so on. The lowest level of the hierarchy contains the bottom time series (b1, b2, b3, b4 in this case), while all the remaining time series are referred to as upper time series. The base forecasts are the univariate forecasts produced independently for each time series; they are incoherent, i.e., they do not satisfy the summing constraints. Reconciliation adjusts the base forecasts to make them coherent. Coherence is hence the main goal of reconciliation; however, usually reconciled forecasts are also more accurate than the base forecasts.
>
> **Experiments**. All the tested reconciliation methods are indeed compared against "normal forecasting method" without reconciliation (base forecasts using the reconciliation terminology). In Table 2 we report the skill scores, i.e., the percentage improvement of the indicator compared to the base forecasts (the skill score is defined on page 7, on the left). Positive skill scores indicate an improvement compared to the base forecasts. Besides our methods, we only test the Gaussian and truncated Gaussian methods; there is currently no specific method for the reconciliation of mixed hierarchies.
>
> **Figures**. We will give a more detailed explanation in the paper. For example, in Fig. 2, we show the effect of the reconciliation on the base distribution of the upper time series of a minimal hierarchy. In Fig. 3, we compare the one-step ahead base forecast distribution with the reconciled distributions via mix-cond and TD-cond on a time series taken from M5.
>
> **Table 2**. In Table 2, we report the already mentioned skill scores. Mix-cond is the best method on moderately sized datasets (such as syph and syph-small), while TD-cond is the best method on large datasets (such as M5). Remarkably, Gaussian and truncated Gaussian methods have negative scores in most cases, i.e., the forecasts reconciled by these methods are less accurate than the base forecasts. We will expand the "Results" paragraph of Sect. 5 to clarify these points.
>
> **Levels**. We refer to all the aggregated levels as “upper” (not only the top one), but they are not merged into a single level, as we also consider the constraints between different upper levels. For instance, in the M5 store hierarchy (Fig. 5), “store”, “category”, and “department” are the upper levels, while “items” is the bottom level.

---

### Meta-Review · Area_Chair_mm2A · 2024-04-17

This paper proposed two methods for probabilistic reconcilation for mixed-type hierarchical time series. Mixed-type hierarchical reconcilation is an important and less studied problem in time series forecasting. The two methods are mixed-conditioning and top-down conditioning.

The reviewers have reached a consensus on accepting the paper. The reviewers pointed out that this paper tackles an important and less studied problem with decent results. The method is explained in a logical manner. Experiments are well-designed. The paper is well-written. On the negative side, the experimental evaluation can be expanded.